

# Comparison of VOC measurements made by PTR-MS, Adsorbent Tube/GC-FID-MS and DNPH-derivatization/HPLC during the Sydney Particle Study, 2012: a contribution to the assessment of uncertainty in current atmospheric VOC measurements

Erin Dunne[1], Ian E. Galbally[1], Min Cheng[1], Paul Selleck[1], Suzie B. Molloy[1], Sarah J. Lawson[1]

[1]CSIRO Oceans and Atmosphere, Aspendale, 3195, Australia

*Correspondence to*: Erin Dunne (erin.dunne@csiro.au)

**Abstract.** Understanding uncertainty is essential for utilizing atmospheric VOC measurements in robust ways to develop atmospheric science. This study describes an inter-comparison of the VOC data, and the derived uncertainty estimates,

measured with three independent techniques (PTR-MS, AT-GC-FID and DNPH-HPLC) during the Sydney Particle Study campaigns in 2012. The compounds and compound classes compared, based on objective selection criteria from the available data, were: benzene, toluene, $C_8$ aromatics, isoprene, formaldehyde, acetaldehyde and acetone. Bottom-up uncertainty analyses were undertaken for each compound and each measurement system. Top-down uncertainties were quantified via the inter-comparisons. Four metrics were used for the inter-comparisons: the slope and intercept as determined by reduced major axis

regression, the correlation, and the root mean standard deviation of the observation from the regression line. In all seven comparisons the correlations between independent measurement techniques were high with $R^2$ values of median 0.93 (range: 0.72 - 0.98) and small root mean standard deviations of the observations from the regression line with a median of 0.13 (range: 0.04 - 0.23 ppb). These results give a high degree of confidence that for each comparison the response of the two independent techniques are dominated by the same constituents. The slope and intercept as determined by reduced major axis regression

gives a different story. The slopes varied considerably with a median of 1.23 and range 1.08 to 2.03. The intercepts varied with a median of 0.02 and range -0.07 to 0.31 ppb. An ideal comparison would give a slope of 1.00 and an intercept of zero.

This analysis identified some poorly understood and poorly quantified sources of uncertainty in the measurement techniques including: the contributions of non-target compounds to the measurement of the target compound for benzene, toluene and isoprene by PTR-MS; and, the under-reporting of formaldehyde, acetaldehyde and acetone by the DNPH technique. As well

as these, this study has identified a specific interference of liquid water with acetone measurements by the DNPH technique. These relationships reported for Sydney 2012 were incorporated into a larger analysis with 61 other published inter-comparison studies for the same compounds. Overall for the light aromatics, isoprene and the $C_1 - C_3$ carbonyls the uncertainty in a set of measurements varies by a factor of between 1.5 and two. These uncertainties (~ 50%) are significantly higher than uncertainties estimated using standard propagation of error methods, which in this case were ~ 22% or less, and are the result of the presence

of poorly understood or neglected processes that affect the measurement and its uncertainty. The uncertainties in VOC measurements identified here should be considered when: assessing the reliability of VOC measurements from individual instruments; when utilising VOC data to constrain and inform air quality and climate models; when using VOC observations for human exposure studies; and, when comparing ambient VOC data with satellite retrievals.

**Keywords:** Uncertainty, VOC, PTR-MS, GC-FID, DNPH, inter-comparison, urban air.

**1 Introduction**

Volatile organic compounds (VOCs) in the atmosphere have important roles in processes leading to formation of ozone and secondary organic aerosol (SOA), and quantitative measurements of VOCs are important for source reconciliation, verification



of atmospheric models and exposure assessment. While atmospheric VOC measurements commenced around 60 years ago, measurement techniques are still rapidly evolving and the uncertainties associated with these measurements are often poorly understood. Assessment of uncertainty for VOC measurement techniques by standard methods (Harris 2003; JCGM 2008) often underestimates what happens in practice because of the presence of poorly understood or neglected processes that affect

5 the measurement and its uncertainty. However comparison of independent techniques for measuring individual VOCs provides a more critical test of uncertainties. Inter-comparison of independent techniques and their quantification of measurement uncertainty, can collectively contribute significantly to the tasks of validation of a wider range of new knowledge, particularly where atmospheric VOC observations are used to validate VOC emissions inventories, air chemistry models and human exposure to air toxics.

Uncertainty in measurements of atmospheric constituents, including VOCs, can arise from four components of the measurement process:

- The pre-treatment of the sample (e.g. in the inlet or adsorption, storage and desorption on a cartridge)
- The matrix in which the sample is presented to the detector (e.g. in nitrogen, helium, air, or some complex mixture)

15 - The presence of interfering compounds in the sample (e.g. co-eluting in chromatography or isobaric compounds in mass spectrometry)
- The instrument calibration (e.g. calibration standards used, linearity of detector response)

There are two distinct methods of determining these uncertainties in VOC measurements. In the first approach, one can examine the individual components of a single measurement technique and assess the uncertainty of each and combine these

20 to get a total uncertainty for that method as described in the Guide to Expression of Uncertainty (JCGM 2008). With this approach, one question always remains: were any sources of uncertainty overlooked? The other method is to make multiple paired measurements with different measurement techniques, of either synthetic VOC mixtures in cylinders or from air in chambers, or ambient air, and determine the uncertainty from the resulting paired measurements. This again only captures a partial contribution to the uncertainty, but it is particularly effective in identifying the presence of unknown sources of

25 uncertainty and complements the first approach. While both approaches were undertaken in this study, it is the later approach that primarily is examined here.

Three independent VOC measurement systems were employed in the study presented here: continuous measurements by proton transfer reaction mass spectrometry (PTR-MS); integrated 5-10 h samples on VOC adsorbent tubes with subsequent offline analysis by GC-FID-MS based on USEPA Method TO 17 (USEPA, 1999a); and integrated 5-10 h samples on 2,4-

30 dintro-phenyl-hydrazine (DNPH) cartridges with subsequent offline analysis by HPLC based on US EPA Method TO 11A (USEPA, 1999b).While notable examples exist e.g. Kajos et al (2015) there is no widely accepted procedure for assessing uncertainty in PTR-MS measurements. Furthermore, while TO 17 and TO 11A provide quality control criteria they do not provide a procedure for systematic uncertainty analysis.

The Sydney Particle Study (SPS), was an intensive field experiment designed to provide a detailed characterisation of the

35 chemical and aerosol composition of the urban atmosphere in Sydney, Australia, in summer 2011 and autumn 2012 (Cope et al., 2014). Sydney is Australia's largest city (population ~4.3 million) and occasionally (~3 days yr$^{-1}$) experiences exceedances of minimum air quality standards for ozone and particulate matter (PM 2.5) (OEH, 2015). In Sydney, the VOCs present were dominated by those from biogenic sources, motor vehicles, bushfires and domestic wood heating (CSIRO 2008, Cope et al 2014).

40 During the second SPS campaign, SPS 2, in autumn from 15$^{th}$ April – 12$^{th}$ May 2012, three independent VOC measurement systems were deployed: continuous measurements by proton transfer reaction mass spectrometry (PTR-MS), and integrated 5-10 h samples on both VOC adsorbent tubes and 2,4-dintro-phenyl-hydrazine (DNPH) cartridges, with subsequent off-line analysis. The measurement site was approximately 1000 km from the parent laboratory, where the equipment was transported



and assembled before the study, therefore the results are typical of normal operating conditions for these instruments rather than that of a specially selected intensive inter-comparison study.

The compounds selected for discussion in the proceeding analysis are a subset of the species measured by the PTR-MS, AT-VOC and DNPH techniques in SPS 2. For the full results of the PTR-MS, AT-VOC and DNPH analysis from SPS 2 the reader
is referred to Keywood et al. (2016).

We present quantitative comparisons of concentrations of VOCs including (a) $C_6$-$C_8$ aromatic compounds and isoprene by PTR-MS and integrated VOC adsorbent tube measurements with subsequent GC-FID-MS analyses and (b) formaldehyde, acetaldehyde and acetone by PTR-MS VOC measurements and carbonyl compounds sampling onto DNPH cartridges followed by HPLC analysis. The results are discussed with regard to the primary responses, interfering species, standard uncertainty
analyses and the limitations of the methods.

The results from this study are compared with other inter-comparison data from the scientific literature and some conclusions about the uncertainty in current VOC measurements presented.

## 2 Methods

### 2.1 Measurement Site and set-up

The sampling site (33.802° S, 150.998° E) was located in the suburb of Westmead, in the grounds of a psychiatric hospital at a position greater than 500m from major roads. Measurements were conducted over the period 15$^{th}$ April – 13$^{th}$ May 2012. The PTR-MS and the sampling apparatus for both the VOC adsorbent tubes and the DNPH cartridges were located in a demountable building surrounded by a grass covered area with occasional trees and the nearest buildings were > 20 m away. The main VOC sampling inlet was ~1m above the roofline, consisting of a ~80 mm O.D. glass inlet of ~2m length.

### 2.2 Carbonyl-DNPH derivatization analysed by HPLC

Ambient air was drawn from the main VOC sample inlet via ~ 4 m length of ¼ inch silco-steel tubing into a custom designed automated sampler. The automated sampler is a continuous air sampler with two channels allowing for simultaneous extractive sampling onto VOC adsorbent tubes and DNPH cartridges. Three samples per day (5:00-10:00, 11:00-19:00 and 19:00-5:00)
were collected by the automated sampler which actively drew air through DNPH coated solid silica adsorbent cartridges (Supelco LpDNPH S10, Supelco Pennsylvania, USA), using a constant flow air sampling pump at a set flow rate of 1 L min$^{-1}$. There is a known deterioration, over one or more days, of derivatized DNPH-carbonyl samples at room temperature. Because of this, the compartment housing the DNPH cartridges in the automated sampler was maintained at ~7° C and the cartridges were refrigerated before and after sampling. An ozone scrubber (KI impregnated filter) was placed in front of the DNPH
cartridges.

The method of DNPH-HPLC sampling employed in this study is compatible with USEPA method TO-11A(USEPA, 1999b). Following sampling, the derivatives were eluted from the cartridge in 2.5 mL of acetonitrile (HPLC-grade, Merck) and analysed by high performance liquid chromatography (HPLC) consisting of a Dionex GP40 gradient pump, a Waters 717 autosampler, a Shimadzu System controller SCL-10A VP, a Shimadzu diode array detector (DAD) SPD-M10A VP, a
Shimadzu Column Oven CTO-10AS VP and Shimadzu CLASS-VP chromatography software. Compound separation was performed with two Supelco Supelcosil LC-18 columns in series (5 μm, 4.6 mm I.D., 250 mm length, Part No. 58298). The chromatographic conditions include a flow rate of 1.6 mL min$^{-1}$ and an injection volume of 25 μL, and the DAD was operated in the 220 – 520 nm wavelength range with 360 nm used for mono-carbonyl quantification. The peaks were separated by gradient elution with an initial mobile phase of 64% acetonitrile and 36% deionized water (18.2 ΩM cm, Millipore Milli-Q
Advantage) for 10 min, followed by a linear gradient to 100% acetonitrile for 20 min, and with a column temperature of 30



°C. A certified liquid standard (Supelco Carb Method 1004 DNPH mix 2 C/N 47651-U) containing 30 µg mL$^{-1}$ of each derivatised carbonyl was diluted 1:25 in a volumetric flask. This prepared standard was then used to perform a 4 point calibration (0.15, 0.30, 0.6 and 1.2 µg mL$^{-1}$). Further details of the DNPH method can be found in Lawson et al. (2008b).

**2.3 VOC adsorbent tubes analysed by GC-FID-MS (AT-VOC)**

In SPS 2, three samples per day (5:00-10:00, 11:00-19:00 and 19:00-5:00) were collected by the automated sampler which actively drew air through two multi-adsorbent tubes in series (Markes Carbograph / Carbopack X) using a constant flow air sampling pump at a set flow rate of 20 mL min$^{-1}$. The adsorbent tubes were analysed by a PerkinElmer TurboMatrix$^{TM}$ 650 ATD (Automated Thermal Desorber) and a Hewlett Packard 6890A gas chromatograph (GC) equipped with flame ionization detection (FID) and a mass spectrometer (MS).

Further details of this method can be found in Cheng et al. (2008a). The method of adsorbent tube VOC sampling (AT-VOC) and analysis employed in this study was compatible with ISO16017-1:2000 and in accordance with USEPA Compendium method TO-17(USEPA, 1999a).

A series of certified gas standards including a: BTEX standard (benzene, toluene, ethylbenzene and xylenes) (Air Liquide-Scott Specialty Gases: Longmont CO USA); a BTEX plus isoprene standard (NPL, Middlesex, UK); a TO-15 standard (Air

Liquide, Plumsteadville PA USA); a PAMS gas standard (Spectra Gases, Linde NJ USA) and were used to the calibrate the GC-FID-MS. The calibration was done via an injection of the calibration gas onto an adsorption tube using a fixed volume temperature stabilised loop for standards with > 2 ppm individual VOCs and via sampling a known volume of calibration gas onto an adsorption tube using a calibrated mass flow controller for standards with < 2 ppm individual VOCs.

**2.4 Proton Transfer Reaction – Mass Spectrometry (PTR-MS)**

A flow of 1.5 L min$^{-1}$ of ambient air was drawn off the main VOC inlet line via a second ~ 4 m length of ¼ inch O.D. silco-steel tubing by a constant flow sampling pump through the PTR-MS auxiliary system and the PTR-MS sampled 300 mL min$^{-1}$ from the auxiliary system.

In SPS 2 a commercially built PTR-MS (Ionicon Analytik, GmbH, Innsbruck Austria) was utilised for continuous VOC

measurements. For a detailed description of PTR-MS the reader is referred to (Ellis and Mayhew 2014). Briefly, the instrument consists of hollow cathode ion source where reagent ions were generated, a drift tube where the reagent ions and the sample were mixed and chemical ionisation reactions occurred between the reagent and the analytes, and a quadrupole mass spectrometer (Balzers QMG422) with a secondary electron multiplier (SEM) operating in pulse counting mode, for sorting and detecting reagent and product ions.

The drift tube was operated at 60° C, and an applied voltage of 445 V and a pressure of 2.16 mbar. The PTR-MS quadrupole continuously scanned 181 masses between 14 and 200 amu with a dwell time for a single mass (m/z) of 1 s, generating a full mass scan approximately every 3 min (20 data points h$^{-1}$ m/z$^{-1}$).

The PTR-MS operated with the aid of custom built auxiliary equipment that regulated the flow of air in the sample inlet and controlled whether the PTR-MS was sampling ambient or zero air or calibration gas. The timing and duration of zero,

calibration and ambient measurement for SPS 2 are detailed in Table 1. Zero readings were made by diverting ambient air through a zero furnace (350° C) with a platinum wool catalyst that destroyed VOCs in the air before entering the PTR-MS. This zero air had the same mole fractions of $H_2O$ and $CO_2$ as the ambient air being sampled, neglecting minor contributions from the oxidation of the VOCs present.

All PTR-MS ion signals from calibration and ambient measurements referred to in this study were background corrected.

The minimum detectable limit for each m/z scanned by the PTR-MS was determined from the scatter in the zero measurements using the principles of ISO6879 (ISO, 1995). The MDL for a single measurement was set at the 95$^{th}$ percentile of the deviations



about the mean zero. The PTR-MS was calibrated daily for 30 min. For each calibration measurement a set flow of 10-20 mL min$^{-1}$ of the calibration standard was diluted in a flow 1500 mL min$^{-1}$ of ambient air that had been passed through the zero furnace.  The empirically derived calibration factors for the 7 compounds of interest to this study, which were included in the calibration standards are listed in the Table 2. The scatter (±1 σ) in the calibration measurements was ~10% (range 6 – 21%).

The PTR-MS was calibrated with three certified gas standards containing in total 20 VOC species. These certified gas standards were supplied by Apel-Reimer Environmental Inc (Broomfield CO, USA), and Air Liquide–Scott Specialty Gases (Plumsteadville, PA USA). The stated accuracy for each component in the standards was ± 5%.

The gravimetrically prepared Apel Reimer standard used to calibrate the PTR-MS, contained benzene, toluene, and m-xylene, among other components. This standard was also analysed with the GC-FID-MS against a certified BTEX gas standard (Air

Liquide–Scott Specialty Gases). The FID response factors for the 2 standards differed by 5 – 9% (BTEX/Apel Reimer Ratios: benzene 0.95; toluene 0.95 and m-xylene 0.91) and we can conclude that the PTR-MS and GC-FID-MS calibrations were compatible within these limits.

**2.6 Criteria for measurement comparisons**

While a number of compounds were measured by both the PTR-MS and AT-VOC or DNPH techniques, only compounds

whose data met the following criteria were retained for the analysis:

1.  Each PTR-MS sample had an ambient data acquisition period that was > 90% of the integrated sampling period of the AT-VOC or DNPH for each sample

2.  Each compound known to substantially contribute to a given m/z signal in PTR-MS measurements of the atmosphere, were also measured in the AT-VOC and/ or DNPH samples

3.  An empirical calibration from measurements of a certified standard containing the compound/s of interest was available for both techniques being compared

4.  The ratio of the median/MDL was > 5 for both datasets for the compounds being compared (Table 3)

The averaging periods used to merge the PTR-MS, AT-VOC and DNPH data from SPS 2 are listed in Table 1. Three DNPH cartridges and three pairs of VOC adsorbent tubes were collected daily: a 5 h sample collected in the morning (5:00-10:00);

an 8 h sample collected in the afternoon (11:00-19:00) and a 10 h sample collected over night (19:00-5:00). Three averages were determined from PTR-MS data that corresponded, with the three integrated sampling periods listed above (see Table 1). Of the range of compounds measured by each of the three VOC measurement systems (PTR-MS, AT-VOC, DNPH), the data for seven compounds/compound groups satisfied criteria 1, 2, 3 and 4 for inclusion in the inter-comparison presented here; they were benzene, toluene, the C$_8$ aromatics, isoprene, formaldehyde, acetaldehyde and acetone.

**2.7 Uncertainty in VOC measurements and inter-comparisons**

There were two methods of determining uncertainties in VOC measurements assessed in this study. The first approach, the bottom up method, examined the individual components of a single measurement technique, assessed the uncertainty of each and combined these to get a total uncertainty for that method (Harris, 2003; JCGM, 2008b).The uncertainty analysis proceeded via the mathematical model, here called the measurement equation, for the measurement as described in the Guide to

Expression of Uncertainty in Measurement (JCGM, 2008b). Details of the uncertainty analysis procedure for each of the selected compounds and for measurement technique are described in the Supplement 1. All uncertainties in this analysis are expanded uncertainties with a coverage factor k = 2. The associated level of confidence of the uncertainty interval is typically 95%.

In the second approach to assessing uncertainty, the top-down method, we evaluated the systematic difference between two

methods by evaluating the slope and intercept of a linear regression between two sets of paired simultaneous measurements. We evaluate random deviations of individual measurements as the root mean square of the orthogonal distance between the





location of the pair of observations (x,y) and the regression line for the whole data set, here referred to as the root mean square of the deviations (RMSD) (Harris, 2003).

When comparing two observational datasets reduced major axis (RMA) regression is preferable to simple least squares linear regression because the analysis is not between an independent and dependent variable, and RMA accounts for random

measurement error on both the x- and y- variables, rather than only the y-variable (Kermack and Haldane, 1950; Ayers, 2001). Contributions to the uncertainty of these measurements that are not included in the bottom-up analyses but are apparent from the top-down analyses are discussed. These contributions are described as poorly understood and poorly quantified processes that do not occur in the measurement equation.  Some examples of these for PTR-MS and DNPH are identified. None were immediately apparent for AT-VOC.

The results this inter-comparison are compared with similar published studies from the scientific literature and some conclusions about the uncertainty in current VOC measurements presented. The other studies examined were published in the peer-reviewed literature, all employed PTR-MS as one of the instruments being compared, only results of ambient air studies were included (direct measurements of VOC emission sources such as biomass burning plumes were excluded) and in all comparisons both instruments were calibrated for the species of interest.

**3. Results and Discussion**

Seven sets of inter-comparisons matched the criteria presented in section 2.6. These were:

- Benzene, toluene, the sum of the $C_8$ aromatics and isoprene measured by both the PTR-MS and the AT-VOC techniques in SPS 2.
- Formaldehyde, acetaldehyde and acetone measured by both the PTR-MS and the DNPH techniques in SPS 2.

For simplicity, the subsequent text is organised around the names of the most common compound/s occurring in the instrument response, while the discussion recognizes that other interfering or co-eluting compounds can be contributing to the instrument response.

The MDL, summary statistics (25th percentile, median, 75th percentile) and the median/MDL for each compound are presented in Table 3.

The uncertainty associated with measurement of these VOCs is evaluated via the methods in the Guide to Expression of Uncertainty in Measurement (JCGM 2008) and presented in the Supplementary Material. While there is some overlap between the observed uncertainty and the calculated measurement uncertainty, they also include distinct components. The observed uncertainty of a set of atmospheric VOC measurements includes a component due to atmospheric variability that is not included in the calculated uncertainty. The calculated measurement uncertainty includes a component due to uncertainty in the

calibration standards, which does not occur in the observed variability of atmospheric measurements which are measured against one reference standard.

Here we analyse whether the sets of simultaneous measurements of VOCs by two different methods have uncertainties such that their mean values ± the measurement uncertainties overlap within the 95% confidence limit or not. Table 4 shows that for benzene, isoprene, acetaldehyde and acetone the mean values do not overlap within the 95% confidence limits. Whereas, for

toluene, xylenes and formaldehyde the mean values do overlap within the 95% confidence limits.

**3.1 Inter-comparison of PTR-MS and AT-VOC samples analysed by GC-FID-MS**

The inter-comparisons for benzene, toluene, the sum of the $C_8$ aromatics and isoprene measured by both the PTR-MS and the AT-VOC techniques are presented in Table 5 as the slope and intercept of the RMA regression analysis, correlation ($R^2$) and the RMSD for each compound and scatterplots of the data are presented in Figure 1a - e.





### 3.1.1 Benzene

In PTR-MS benzene is detected at m/z 79. Reduced major axis regression analysis between the PTR-MS data for m/z 79 and the AT-VOC benzene data yielded a slope of $1.47 \pm 0.04$, an intercept of $0.02 \pm 0.00$ppbv, $R^2 = 0.96$ (Figure 1a). The high $R^2$ value and small RMSD = 0.04 ppbv (RMSD/Median = 8%) (Table 5) indicates the AT-VOC and PTR-MS were both

responding to benzene. The comparisons presented in Table 4 indicate a significant difference at the 95% confidence limit between the mean values measured by each instrument.

It is possible the slope of ~ 1.5 was a result of contributions to the PTR-MS signal at m/z 79 from compounds other than benzene, such as fragment ions from ethylbenzene, propyl- and isopropyl-benzene, butyl- and isobutyl-benzene which can potentially contribute to the signal at m/z 79 (Warneke et al., 2003). In addition, an unknown $CH_2O_4H^+$ ion signal was detected

at m/z 79 by high resolution PTR- time of flight (ToF) MS in a rural atmosphere (Park et al., 2013).

In a separate study, the PTR-MS was exposed to a certified gas standard containing roughly equivalent VMRs of benzene, ethylbenzene, propyl- and isopropyl-benzene, among other components. The signal at m/z 79 was 41% higher than in measurements of a standard containing benzene but not ethylbenzene, propyl- and isopropyl-benzene also tested.

To evaluate the potential contribution of ethylbenzene , isopropyl- and propyl-benzene to the PTR-MS signal at m/z 79 in

SPS2 , the m/z 79 data was compared with the AT-VOC data for benzene corrected ($[Benz]_{corr}$), with the addition of ethylbenzene (EtBenz), propylbenzene (PrBenz) and isopropylbenzene (iPrBenz) where the data has been corrected to account for differences in the reaction rate coefficients (k) ($10^9$ $cm^3$ $sec^{-1}$) with the $H_3O^+$ chemical ionization reagent in the PTR-MS relative to benzene, and their branching ratios (BR) to m/z 79 determined from PTR-MS reference spectra:

$$[Benz]_{corr} = [Benz] + \left([EtBenz] \times \frac{k_{EtBenz}}{k_{Benz}} \times BR_{EtBenz}\right) + \left([PrBenz] \times \frac{k_{PrBenz}}{k_{Benz}} \times BR_{PrBenz}\right)$$

$$+ \left([iPRBenz] \times \frac{k_{iPrBenz}}{k_{Benz}} \times BR_{iPrBenz}\right)$$

Equation 1a

$$[Benz]_{corr} = [Benz] + \left([EtBenz] \times \frac{2.22}{1.93} \times 0.18\right) + \left([PrBenz] \times \frac{2.44}{1.93} \times 0.09\right) + \left([iPRBenz] \times \frac{2.44}{1.93} \times 0.56\right)$$

Equation 1b

where, branching ratio values were taken from Gueneron et al. (2015) and reaction rate values were taken from Cappellin et

al (2010) Supplementary material. Ethylbenzene was measured in the AT-VOC samples however, propyl- and isopropyl benzene were not. In the absence of better data, we drew upon a dataset where the concentrations of 81 hourly samples were taken in 5 urban locations during the day (7:00 – 17:00) in March in Sydney 2006 (Azzi et al., 2007).The linear relationships (y-intercept = 0) observed between 1,3,5 trimethylbenzene and propyl- and isopropylbenzene from the 2006 study, (Slopes 0.50, 0.14, $R^2$ 0.87, 0.55 respectively), were used to interpolate propyl- and isopropylbenzene concentrations from the 1,3,5

trimethylbenzene concentrations observed in the AT-VOC samples in this study.

The slope of the RMA regression between the PTR-MS and the corrected AT-VOC data improved moderately to 1.27 ($R^2$ = 0.97), indicating $C_8$ and $C_9$ aromatics made a measurable contribution to the PTR-MS signal at m/z 79 in this study. The degree of interference will vary with the relative concentrations of higher aromatics to benzene in the atmosphere being studied. As the higher aromatics have shorter atmospheric lifetimes than benzene, the interference will vary with ageing of an air mass.

The quantitative agreement between the measurements of benzene by PTR-MS and AT-VOC in this study was poorer than those reported in similar real-world inter-comparisons most of which have reported slopes between 0.8 and 1.2 shown graphically in Figure 2 (Warneke et al., 2001b; de Gouw et al., 2003; Kato et al., 2004; Kuster et al., 2004; Jobson et al., 2005; Rogers et al., 2006; de Gouw and Warneke, 2007; Kaser et al., 2013; Wang et al., 2014; Kajos et al., 2015; Cui et al., 2016).

In summary, a comparison between the measurements of benzene by PTR-MS and the AT-VOC technique indicates a

significant difference in the measured concentrations which may vary according to the relative contribution of higher aromatics





in different atmospheres. The influence of higher aromatics on benzene measurements by PTR-MS is identified as a poorly understood and poorly quantified uncertainty.

### 3.1.2 Toluene

In PTR-MS the signal at m/z 93 is attributed to toluene. RMA regression analysis between the PTR-MS data at m/z 93 and the

AT-VOC data yielded a slope of $1.25 \pm 0.02$, intercept = $-0.03 \pm 0.00$ ppbv, $R^2 = 0.97$ (Figure 1b). The RMSD was 0.11 ppb which was only 5% of the median PTR-MS value (Table 5). The comparisons presented in Table 4 indicate no difference between mean values measured by each instrument at the 95% confidence limit.

The slope > 1 may be a result of contributions to the PTR-MS signal at m/z 93 from compounds other than toluene. These include α- and β-pinene, p-cymene, and several $C_9$ aromatics (ethyltoluenes, 1,2,3-trimethylbenzene), all of which are known

to produce fragment ions at m/z 93 in PTR-MS (Warneke et al., 2003; Malekinia et al., 2007; Ambrose et al., 2010; Gueneron et al., 2015).

These potential interferent compounds, with the exception of p-ethyltoluene and 1,2,3-trimethylbenzene, were measured in the AT-VOC samples. To evaluate the contribution of these compounds to the PTR-MS signal at m/z 93 in this study, we used an analogous correction procedure for the AT-VOC data to that outlined in the previous section (Equation 1):

$$[Tol]_{corr} = [Tol] + \left([\alpha\ pine] \times \frac{2.37}{2.08} \times 0.07\right) + \left([\beta\ pine] \times \frac{2.46}{2.08} \times 0.07\right) + \left([p\ cym] \times \frac{2.50}{2.08} \times 0.66\right)$$
$$+ \left([m\ \&o\ EtTol] \times \frac{2.40}{2.08} \times 0.03\right)$$

Equation 2

Where literature values were used for the reaction rates (Cappellin et al., 2010) and branching ratios (Warneke et al., 2003; Malekinia et al., 2007; Gueneron et al., 2015). This correction had a minor impact on the slope of the RMA regression (slope

= $1.21 \pm 0.02$, intercept = $-0.03 \pm 0.00$, $R^2 = 0.98$) which is close to the criteria for quantitative agreement prescribed for this study.

With the exception of two of studies (Kato et al., 2004; Kajos et al., 2015) previous inter-comparisons between toluene measurements by PTR-MS and GC techniques have reported slopes of $0.8 - 1.2$ and generally good correlations ($R^2 > 0.75$) (Figure 2.) (Warneke et al., 2001a; de Gouw et al., 2003; Kuster et al., 2004; de Gouw and Warneke, 2007; Kaser et al., 2013;

Wang et al., 2014; Kajos et al., 2015; Cui et al., 2016).

In summary, a comparison between the measurements of toluene by PTR-MS and the AT-VOC technique indicates that there was not a significant difference in the measured concentrations. There may be some residual unquantified interference with the PTR-MS toluene measurement.

### 3.1.3 $C_8$ aromatics

In PTR-MS, the signal at m/z 107 is commonly regarded as a measure of the sum of the $C_8$ aromatic isomers (m-, p-, o- xylenes and ethylbenzene) with possible minor contributions from benzaldehyde (de Gouw and Warneke, 2007). Due to fragmentation in the PTR-MS ~ 80% of the ethylbenzene ion signal occurs at m/z 107 (Gueneron et al., 2015).To account for the contribution of all of these compounds to the PTR-MS signal at m/z 107 in this study, we correct the AT-VOC data to an equivalent quantity. The correction procedure for the AT-VOC data is analogous to that outlined in section 3.1.1 (Equation 1):

$$[C_8\ Aromatics]_{corr} = sum[Xyl] + \left([EtBenz] \times \frac{2.22}{2.26} \times 0.8\right) + \left([BenzAld] \times \frac{3.7}{2.26}\right)$$

Equation 3

Literature values were used for the reaction rates and branching ratios (Spanel et al., 1997; Cappellin et al., 2010; Gueneron et al., 2015). RMA regression yielded a slope of $1.19 \pm 0.02$, intercept $-0.03 \pm 0.01$ ppbv, $R^2 = 0.98$ (Figure 1c).The RMSD of



0.09 ppbv was only 7% of the median PTR-MS value (Table 5). The comparisons presented in Table 4 indicate that the mean values reported by each instrument agree within 95% confidence limits.

The results reported here are in-line with many previous intercomparison studies that have reported good quantitative agreement, within ± 20% ($R^2 > 0.85$) between PTR-MS and GC techniques for the measurement of the sum of the $C_8$ aromatics

(Warneke et al., 2001a; Kuster et al., 2004; Jobson et al., 2005; Rogers et al., 2006; de Gouw and Warneke, 2007; Wang et al., 2014; Cui et al., 2016). However slopes as low as 0.6 (Kato et al., 2004) and as high as 3.2 (de Gouw et al., 2003) have been reported with the discrepancy in both cases attributable to calibration inaccuracies.

In summary, a comparison between the measurements of $C_8$ aromatics by PTR-MS and the AT-VOC technique indicates that there was not a significant difference in the measured concentrations. There may be some residual unquantified interference

with the PTR-MS $C_8$ aromatic measurement.

### 3.1.4 Isoprene

In measurements of the atmosphere the PTR-MS signal at m/z 69 is attributed to isoprene. The RMA regression analysis between the PTR-MS and AT-VOC data for isoprene yielded a slope of 1.23 ± 0.07, intercept = 0.31 ± 0.10 ppbv, $R^2 = 0.75$ (Figure 1d). The lower $R^2$ and higher RMSD of 0.13 ppbv (Table 5) observed for isoprene indicate the two instruments may

not have been responding to entirely the same compounds (Figure 1d). The comparisons presented in Table 4 indicate a significant difference at the 95% confidence limit between the mean values measured by each instrument.

Isoprene emissions are dominated by biogenic sources and are strongly light and temperature dependent with maxima in the afternoon. For SPS 2 when only the afternoon data were considered, closer agreement was observed between the PTR-MS and AT-VOC data for isoprene attributed to a 0.2 ppb lower intercept (0.11 ± 0.10 ppb) and significantly higher $R^2$ of 0.93 (slope

= 1.19 ± 0.06, RMSD = 0.12 ppbv) (Figure 1e).

The significant offsets observed in the PTR-MS data of ~0.1ppb during the afternoon, and ~0.3 ppb during the morning and night, were most likely due to contributions from compounds other than isoprene to the PTR-MS signal at m/z 69. Park et al. (2013) observed three peaks at m/z 69 in high-resolution PTR-ToF spectra in a rural area: $C_3H_2O_2H^+$ (~10%), $C_4H_4OH^+$ (~14%), and $C_5H_8H^+$ (~75%). GC-PTR-MS analysis has also shown multiple other species can contribute to m/z 69, specifically 2-,

and 3- methylbutanal, 1-penten-3-ol in urban air (de Gouw et al., 2003); furan in air masses impacted by biomass burning (Christian et al., 2004); and, 2-methyl-3-buten-2-ol in air masses impacted by emissions from pine trees (Karl et al., 2012).

The results reported here are consistent with previous inter-comparisons studies between PTR-MS and GC techniques which have reported slopes of 0.79 – 2.15 often with significant (up to 0.39 ppb) offsets, shown graphically in Figure 2 (de Gouw et al., 2003; Kato et al., 2004; Kuster et al., 2004; Wang et al., 2004; de Gouw and Warneke, 2007; Kaser et al., 2013).Ne

In summary, a comparison between the measurements of isoprene by PTR-MS and the AT-VOC technique indicates a significant difference in the measured concentrations which may vary according to the relative contribution of other species that contribute to the PTR-MS signal at m/z 69 particularly at night. The influence of these compounds on measurements of isoprene by PTR-MS is a poorly understood and poorly quantified uncertainty.

### 3.2 Intercomparison of PTR-MS with DNPH derivatization-HPLC

In the following section, the inter-comparisons for formaldehyde, acetaldehyde and acetone measured by both the PTR-MS and the DNPH-HPLC techniques in SPS 2 will be discussed in turn. The MDL, summary statistics (25th percentile, median, 75th percentile) and the median/MDL values for the PTR-MS and DNPH data for each compound are presented in Table 3. The results of the analysis of measurement uncertainty are presented din Table 4.

The results of the RMA regression analysis and the RMSD for each compound are summarized in Table 5. Scatterplots of the

comparisons for the three carbonyl compounds are presented in Figure 1f - h.





As part of this analysis, we have identified a loss process in the DNPH method due to condensation of $H_2O$ in the cartridges. To explain this loss, some detail of the measurement technique is necessary. The derivatized carbonyl compounds on the DNPH cartridge samples are extracted with a fixed volume of acetonitrile after air sampling and prior to HPLC analysis. The volume of acetonitrile used in the extraction is determined beforehand and the mass of extract afterwards. The masses

determined afterwards are displayed on the bottom pane of Figure 3. For the period 16/4 – 24/4, the extraction masses (g) were ~10% higher than the volume of acetonitrile added in the extraction and also higher than the extraction masses for other sample and blank cartridges analysed in this study. The additional extraction volume was determined to be a result of condensation of $H_2O$ from the air sampled in these DNPH cartridges during sampling due to the cooling of the DNPH cartridge holder. The presence of liquid water appears to substantially reduce the collection efficiency of acetone with concentrations < MDL as

shown in Figure 3. Due to the presence of condensation in the cartridges, the DNPH data for formaldehyde, acetaldehyde and acetone for the period 16/4 – 24/4 were excluded from this analysis.

### 3.2.1 Formaldehyde

In PTR-MS formaldehyde is detected at m/z 31. The measurement of formaldehyde with PTR-MS is complex as its proton transfer chemical ionization reaction with $H_3O^+$ is close to endothermic and loss via back reaction in humid air is non-negligible

(Hansel et al., 1997; Inomata et al., 2008). In order to account for the water vapour dependence of the PTR-MS response to formaldehyde daily instrument background and calibration measurements were made using zero air that had the same mole fractions of $H_2O$ as the ambient air being sampled. The linear relationship observed between the formaldehyde calibration factors measured daily and the respective water vapour density (g m$^{-3}$) was determined, and a corrected calibration factor was applied to the ambient hourly data based on the ambient water vapour density measured hourly.

RMA regression analysis between the PTR-MS signal at m/z 31 and the formaldehyde in the DNPH-HPLC samples yielded a slope of $1.30 \pm 0.04$, intercept = $-0.07 \pm 0.01$ ppbv, $R^2 = 0.92$, and RMSD = 0.14 ppbv (N = 77). The comparisons presented in Table 4 indicate that the mean values reported by each instrument agree within 95% confidence limits.

To examine any possible effect of liquid water, the analysis was repeated excluding the data 16/4 to 24/4, see Section 3.2.1. The results yielded a slope of $1.20 \pm 0.04$, intercept = $-0.02 \pm 0.02$ ppbv, $R^2 = 0.90$, and RMSD = 0.16 ppbv (N = 53) (Table

5) (Figure 1f). The results indicate a minor but significant effect of liquid water.

The slope of 1.3 may be a result of contributions to the PTR-MS signal at m/z 31 from compounds other than formaldehyde. Inomata et al (2008) described a procedure to correct the m/z 31 ion signal for contributions of methanol, ethanol, and methyl hydroperoxide which are known to produce fragment ions at m/z 31 in PTR-MS. Applying the same correction procedure to the data in this study from Sydney 2012 had a negligible affect (Slope = 1.27).

Previous studies have reported PTR-MS values for formaldehyde that were systematically higher than DNPH-HPLC measurements (Wisthaler et al. 2008, Cui et al. 2016) and higher than DOAS and Hanzstch techniques (Wisthaler et al. 2008, Warneke et al 2011). Other studies report DNPH-HPLC values for formaldehyde that were systematically lower than those reported by other analytical methods (DOAS, FTIR, Hantzsch, TDLAS)(Kleindienst et al., 1988; Lawson et al., 1990; Gilpin et al., 1997; Hak et al., 2005; Wisthaler et al., 2006).

In summary, a comparison between the measurements of formaldehyde by PTR-MS and the AT-VOC technique indicates there was not a significant difference in the measured concentrations.

### 3.2.2 Acetaldehyde

The signal at m/z 45 in PTR-MS spectra is commonly attributed to acetaldehyde. RMA regression analysis between the PTR-MS data for m/z 45 and the acetaldehyde values determined from the DNPH-HPLC samples yielded a slope of $1.47 \pm 0.09$,

intercept = $0.14 \pm 0.02$ ppbv, $R^2 = 0.72$ (Figure 1g), and RMSD = 0.11 ppbv (N = 77) (Table 5).





To examine any possible effect of liquid water, the analysis was repeated excluding the data 16/4 to 24/4, see Section 3.2.1. The results were a slope of $1.43 \pm 0.05$, intercept = $+0.08 \pm 0.01$ ppbv, $R^2 = 0.92$, and RMSD = 0.05 ppbv (N = 54). The results indicate an insignificant effect on slope but a substantial increase in the correlation coefficient and reduction in RMSD by excluding the data indicating liquid water. The comparisons presented in Table 4 indicate a significant difference at the 95%

confidence limit between the mean values measured by each instrument.

A positive bias in PTR-MS measurements of acetaldehyde may result from contributions to the m/z 45 signal from compounds other than acetaldehyde. Due to structural constraints the signal at m/z 45 can be either $C_2H_5O^+$ ions, $HCO_2^+$ and/or $CH_3NO^+$. The contribution from protonated carbon dioxide ($HCO_2^+$) is not relevant here as it is removed by the background correction. Two studies using high resolution PTR-ToF have observed a single peak at m/z 45 consisting of $C_2H_5O^+$ (Park et al 2013,

Warneke et al. 2015). The $C_2H_5O^+$ product ions may result from protonated acetaldehyde, protonated vinyl alcohol, protonated ethylene oxide, or fragment ions from ethylene glycol (Wood et al., 2015), ethanol, (Inomata and Tanimoto, 2009), 2-propanol (Inomata and Tanimoto, 2010), methyl ethyl ketone, methyl glyoxal and methyl isobutyl ketone (Dunne, 2016). None of these compounds were likely to be individually present in sufficient concentrations to account for the discrepancy observed in this study, however the combined effect of numerous compounds yielding m/z 45 product ions cannot be dismissed as a possible

explanation.

In an atmospheric simulation chamber study three PTR-MS instruments reported acetaldehyde values close to the known injected value whereas a DNPH method significantly underestimated (~ 30%) the known chamber concentration (Apel et al., 2008). In a recent comparison in urban air between PTR-MS and DNPH-HPLC, Cui et al (2016) reported a slope of ~ 1 between the two methods but a significant positive offset in the PTR-MS data of 0.83 ppbv and $R^2 = 0.56$.

Herrington et al. (2007) reported the collection efficiency of acetaldehyde on DNPH cartridges declined from ~ 100% for a sampling duration of 6 h to ~ 60% for a sampling duration of 12 h, the reasons for which have not been resolved. As 8 and 10 h sampling durations were used for the DNPH sampling in this study, poor collection efficiencies may have resulted in a negative bias in the DNPH-HPLC measurements of acetaldehyde.

As shown in Figure 2, other real-world inter-comparison studies have reported variable agreement between measurements of

acetaldehyde by PTR-MS and GC methods (slopes $0.87 - 1.7$, intercepts $-0.25 - 0.22$, $R^2$ $0.38 - 0.86$) (de Gouw et al., 2003; Warneke et al., 2011; Kaser et al., 2013; Kajos et al., 2015).

In summary, a comparison between the measurements of acetaldehyde by PTR-MS at m/z 45 and the AT-VOC technique indicates there was a significant difference in the measured concentrations. There may be other species that contribute to the PTR-MS signal at m/z 45, and under-reporting in the DNPH measurement that are poorly understood and poorly quantified

uncertainties.

### 3.2.3 Acetone

In PTR-MS measurements the ion signal at m/z 59 is regarded as a measure of acetone, but may also contain contributions from propanal (de Gouw & Warneke 2007). Glyoxal, another common atmospheric carbonyl compound, would also occur at m/z 59, however tests on prepared gas standards suggest PTR-MS does not detect glyoxal in ambient air samples as the

chemical ionization reaction with $H_3O^+$ is endothermic (Thalman et al., 2015; Dunne, 2016).

The RMA regression analysis between the PTR-MS signal at m/z 59 and the sum of acetone and propanal measured in the DNPH samples yielded a slope of $2.01 \pm 0.14$ ppbv, intercept = $0.21 \pm 0.07$ ppbv, $R^2 = 0.76$ and RMSD = 0.24 ppbv (N = 53) (Figure 1e). The lower $R^2$ and high RMSD (median/ RMSD =38%) (Table 5) suggest the DNPH and /or the PTR-MS measurement of acetone suffered some interference. The comparisons presented in Table 4 indicate a significant difference at

the 95% confidence limit between the mean values measured by each instrument.

Ho et al (2014) identified a significant negative bias in the collection efficiency of acetone on DNPH cartridges that was related to humidity, sample flow rate and sample duration. While Ho et al. (2014) used a similar DNPH cartridge type, these authors



reported 35 – 80% of acetone was lost under similar conditions as those experienced in in this study (RH > 70%, sample flow 1 L min$^{-1}$, sample duration 8 – 10 h). When carbonyls pass through the DNPH sorbent, reactions occur involving the addition of the -NH$_2$ group to the -C=O group to form a reaction intermediate. The reaction between DNPH and ketones occurs at a slower rate than for aldehydes resulting in poorer collection efficiencies for ketones. In the second step of the reaction, the

intermediate loses a water molecule to form the hydrazone derivative. Therefore, when the water mixing ratio is high (i.e. high absolute humidity) loss via the back reaction may be substantial.

Previous published atmospheric and chamber study measurements reported PTR-MS values for acetone that were ~ 30 to > 100% higher than simultaneous DNPH-HPLC measurements (Muller et al., 2006; Apel et al., 2008; Cui et al., 2016). Conversely, generally good agreement has been observed between PTR-MS and GC methods and AP-CIMS (Slopes 0.97 –

1.18, intercept = -0.28 – 0.06, R$^2$ = 0.77 - 0.96) (Figure 3) (Sprung et al., 2001; de Gouw et al., 2003; Kaser et al., 2013; Wang et al., 2014; Kajos et al., 2015).

Overall, the PTR-MS signal at m/z 59 was dominated by acetone. Consistent with previous studies, a significant negative bias was identified in sampling of acetone onto DNPH cartridges and further work is required to determine the performance of DNPH cartridge sampling for quantitative measurements of acetone under real-world conditions. At high humidity, the

formation of condensation in DNPH cartridges must be guarded against as stated in TO-11A (USEPA, 1999b).

In summary, a comparison between the measurements of acetone by PTR-MS at m/z 59 and the AT-VOC technique indicates there was a significant difference in the measured concentrations. There appears to be under-reporting in the DNPH measurement that is a poorly understood and poorly quantified uncertainty.

## 4. Discussion and Conclusions

Inter-comparisons have been made between three independent techniques covering the measurement in the atmosphere of benzene, toluene, C$_8$ aromatics, isoprene, formaldehyde, acetaldehyde and acetone. In all seven comparisons the correlations between independent measurement techniques are high with R$^2$ values of median 0.93, range 0.75 to 0.98 and the root mean standard deviation of the observations from the regression line are small with a median of 0.11 range 0.04 to 0.23 ppb for the comparisons. This gives a high degree of confidence that for each comparison the two independent techniques are responding

to the same constituents.

The slope and intercept as determined by reduced major axis regression gives a different story. The slopes vary considerably with a median on 1.23 and range 1.18 to 2.03. The intercepts vary with a median of 0.02 and range -0.07 to 0.31 ppb. An ideal comparison would give a slope of 1.00 and an intercept of zero. Also an analysis of the measurement uncertainties indicates significant differences between the mean concentrations for benzene, isoprene, acetaldehyde and acetone. The reasons for the

variations in slope include the contributions of non-target compounds to the measurement of the target compound for benzene, toluene and isoprene by PTR-MS and the under-reporting of formaldehyde, acetaldehyde and acetone by the DNPH technique. This study has identified specific issues: (a) with the use of PTR-MS in urban areas at night when interferences from other compounds in isoprene measurements are significant and (b) an interference of liquid water in the sample trap with acetone measurements by the DNPH technique. We note that in this study the PTR-MS always has a larger response than the ATD-

VOC and DNPH-HPLC method. We have reviewed this issue and have concluded that this arises due to the PTR-MS responding to fragments from other compounds as well as the target compounds, and the DNPH-HPLC apparently systematically underreporting.

The relationships reported for Sydney 2012 were incorporated into a larger analysis with 61 other inter-comparison studies for the same compounds, found in the recent scientific literature, see Figure 3. For the whole available set of inter-comparisons,

the R$^2$ has a median 0.83, range 0.28 to 0.98, the slopes has a median of 1.02 and range 0.58 to 2.03 and the intercept has a median of 0.01 and range -0.44 to 1.88 ppb. Based on this compilation we conclude that for the light aromatics, isoprene and





the $C_1 - C_3$ carbonyls the uncertainty in a set of atmospheric measurements with current measurement technology varies by a factor between 1.5 and two. These uncertainties from the inter-comparisons (~50%) are significantly higher than uncertainties estimated using standard propagation of error methods presented in Table 4 of 22% or less. The difference is presumably the result of poorly understood or neglected processes that affect these measurements and their uncertainties.

5    There are two qualifications concerning this overall uncertainty analysis. This analysis in no way indicates what the uncertainty is in measurements of other VOC compounds. A smaller uncertainty has been reported for alkanes (Hoerger et al., 2015).  Similarly, if the emissions and concentrations of a VOC are measured with the same technique, or with techniques that are compared, then the uncertainties associated with an atmospheric mass balance compiled using these measurements may be smaller than the case where different VOC measurement techniques that have not been compared are used.

10  The uncertainties in VOC measurements identified here should be considered when: assessing the reliability of VOC measurements from individual instruments; when utilising VOC data to constrain and inform air quality and climate models; when using VOC observations for human exposure studies; and, when comparing ambient VOC data with satellite retrievals.

### 5.  Acknowledgements

The NSW Environment Trust has provided support for this study through the "Atmospheric Particles in Sydney: model-
15   observation verification study". The NSW Office of Environment and Heritage supported the Sydney Particle Study. We also thank the staff and management of Westmead Hospital for assistance during the field campaigns. This work was initiated when E.D. was undertaking her PhD studies at Monash University. E.D. thanks Monash University for a postgraduate scholarship.



**Table 1. Ambient sampling times from SPS 2 for the PTR-MS , AT-VOC and DNPH-HPLC methods; Zero and calibration times for the PTR-MS.**

| | *Ambient sampling* | | | *Zero* | *Calibration* |
|---|---|---|---|---|---|
| | *Morning* | *Afternoon* | *Night* | | |
| PTR-MS | 5:00 – 10:00 | 11: 00 – 16:45 plus 17:15 – 19:00 | 19:00 – 23:45 plus 0:45 – 5:00 | 23:45 – 0:15 16:45 – 17:15 | 0:15 – 0:45 |
| AT-VOC & DNPH | 5:00 – 10:00 | 11:00 – 19:00 | 19:00 – 5:00 | | |





**Table 2: The PTR-MS calibration factors for each of the VOCs included in this work, normalised to $10^6$ counts per second (cps) of $H_3O^+$ reagent ions per ppb (ncps ppbv$^{-1}$). The uncertainty limits represent ± the relative standard deviation of the mean. N represents the number of 30 min calibration periods used to calculate the sensitivity statistics. The average calibration for formaldehyde is presented in the Table; the ambient data processing for formaldehyde utilized a linear equation Calibration Factor = 16.08 −**
5 **0.232*[$H_2O$], where the water vapour concentration is in g m$^{-3}$.**

|  | MW | m/z | Calibration Factor ncps ppbv$^{-1}$ |
|---|---|---|---|
| Formaldehyde | 30 | 31 | 1.36 ± 21% |
| Acetaldehyde | 44 | 45 | 19.81 ± 6% |
| Acetone | 58 | 59 | 24.02 ± 7% |
| Isoprene | 68 | 69 | 8.84 ± 17% |
| Benzene | 78 | 79 | 17.15 ± 6% |
| Toluene | 92 | 93 | 19.87 ± 6% |
| m-xylene | 106 | 107 | 19.78 ± 8% |
| 1,3,5-trimethylbenzene | 120 | 121 | 17.72 ± 13% |





**Table 3. The MDL and summary statistics (ppb) for the PTR-MS, AT-VOC and DNPH data for each of the 7 compounds selected for this study. Note: the DNPH MDL differs between morning afternoon and night samples due to different sampling times result. For the purposes of this table the DNPH MDLs and Median/ MDLs are quoted as a range.**

|  |  | MDL (ppb) | 25th%ile (ppb) | Median (ppb) | 75th%ile (ppb) | Median/MDL | N |
|---|---|---|---|---|---|---|---|
| Benzene | AT-VOC | 0.005 | 0.20 | 0.36 | 0.69 | 72 | 75 |
|  | PTR-MS | 0.003 | 0.28 | 0.48 | 0.89 | 160 | 75 |
| Toluene | AT-VOC | 0.005 | 0.74 | 1.61 | 2.57 | 322 | 75 |
|  | PTR-MS | 0.003 | 0.89 | 2.01 | 3.02 | 670 | 75 |
| $C_8$ Aromatics | AT-VOC | 0.008 | 0.58 | 1.04 | 1.89 | 130 | 75 |
|  | PTR-MS | 0.003 | 0.68 | 1.33 | 2.23 | 443 | 75 |
| Isoprene | AT-VOC | 0.002 | 0.09 | 0.13 | 0.22 | 65 | 75 |
|  | PTR-MS | 0.014 | 0.35 | 0.51 | 0.80 | 36 | 75 |
| Formaldehyde | DNPH | 0.025 - 0.051 | 0.72 | 0.96 | 1.20 | 18 - 44 | 53 |
|  | PTR-MS | 0.212 | 0.81 | 1.04 | 1.47 | 5 | 53 |
| Acetaldehyde | DNPH | 0.065 - 0.133 | 0.36 | 0.50 | 0.66 | 4 - 7 | 53 |
|  | PTR-MS | 0.024 | 0.58 | 0.80 | 0.98 | 33 | 53 |
| Acetone | DNPH | 0.069 - 0.142 | 0.39 | 0.61 | 0.92 | 5 - 9 | 53 |
|  | PTR-MS | 0.013 | 1.09 | 1.49 | 1.93 | 115 | 53 |





**Table 4. The means and standard deviations of the atmospheric data, the estimated measurement uncertainties of the means (k = 2), see Supplementary Material, the 95% confidence limit of the means (ppb) for the seven compounds measured by PTR-MS, AT-VOC and DNPH and the number of paired observations, N, in this study.**

| | | Mean | SD | Rel. Total Uncertainty of Mean ($k = 2$) | Mean ± Uncertainty ($k = 2$) | N |
|---|---|---|---|---|---|---|
| | | (ppb) | (ppb) | % | (ppb) | |
| Benzene | PTR-MS | 0.59 | 0.38 | 11 | 0.53 – 0.65 | 75 |
| | AT-VOC | 0.45 | 0.30 | 12 | 0.40 – 0.50 | 75 |
| Toluene | PTR-MS | 2.15 | 1.44 | 11 | 1.92 – 2.38 | 75 |
| | AT-VOC | 1.81 | 1.19 | 12 | 1.59 – 2.03 | 75 |
| C$_8$ Aromatics | PTR-MS | 1.49 | 0.99 | 12 | 1.31 – 1.67 | 75 |
| | AT-VOC | 1.28 | 0.83 | 13 | 1.12 – 1.44 | 75 |
| Isoprene | PTR-MS | 0.61 | 0.39 | 19 | 0.50 – 0.72 | 75 |
| | AT-VOC | 0.24 | 0.32 | 7 | 0.22 – 0.26 | 75 |
| Formaldehyde | PTR-MS | 1.27 | 0.71 | 22 | 0.99 – 1.55 | 53 |
| | DNPH | 1.07 | 0.59 | 9 | 0.98 – 1.16 | 53 |
| Acetaldehyde | PTR-MS | 0.84 | 0.40 | 19 | 0.68 – 1.00 | 53 |
| | DNPH | 0.53 | 0.28 | 12 | 0.47 – 0.59 | 53 |
| Acetone | PTR-MS | 1.69 | 1.04 | 22 | 1.32 – 2.06 | 53 |
| | DNPH | 0.74 | 0.52 | 12 | 0.65 – 0.83 | 53 |





**Table 5. The (m), intercepts (b) and correlation coefficients ($R^2$) from the RMA regression analysis between the PTR-MS, AT-VOC and DNPH-HPLC measurements. Also included are the estimates of random measurement uncertainty expressed as RMSD for each**
5 **species and the ratio of the RMSD to the median PTR-MS value expressed as %.**

| m/z | Compound | Slope (m) | Intercept(b) (ppbv) | $R^2$ | RMSD (ppbv) | RMSD/Median | N |
|-----|----------|-----------|---------------------|-------|-------------|-------------|---|
| [PTR-MS] = m × [AT-VOC] + b | | | | | | | |
| 79 | **Benzene** | 1.27 ± 0.03 | 0.02 ± 0.03 | 0.97 | 0.04 | 8 % | 75 |
| 93 | **Toluene** | 1.21 ± 0.02 | -0.03 ± 0.00 | 0.98 | 0.11 | 5 % | 75 |
| 107 | **C$_8$ Aromatics** | 1.19 ± 0.02 | -0.02 ± 0.01 | 0.98 | 0.09 | 7 % | 75 |
| 69 | **Isoprene** | 1.23 ± 0.07 | 0.31 ± 0.10 | 0.75 | 0.13 | 25 % | 75 |
| | (afternoon only) | 1.18 ± 0.06 | 0.11 ± 0.10 | 0.93 | 0.12 | 28% | 26 |
| [PTR-MS] = m × [DNPH-HPLC] + b | | | | | | | |
| 31 | **Formaldehyde** | 1.20 ± 0.04 | -0.02 ± 0.02 | 0.90 | 0.16 | 15 % | 53 |
| 45 | **Acetaldehyde** | 1.43 ± 0.05 | 0.08 ± 0.01 | 0.92 | 0.05 | 6% | 53 |
| 59 | **Acetone** | 2.01 ± 0.14 | 0.21 ± 0.07 | 0.76 | 0.23 | 15% | 53 |







**Figure 1. Intercomparisons of PTR-MS versus AT-VOC and DNPH measurements of selected VOCs in SPS 2 (2012). RMA correlation coefficients ($R^2$) and regression fits are indicated (solid line) ± std error (dashed lines).**





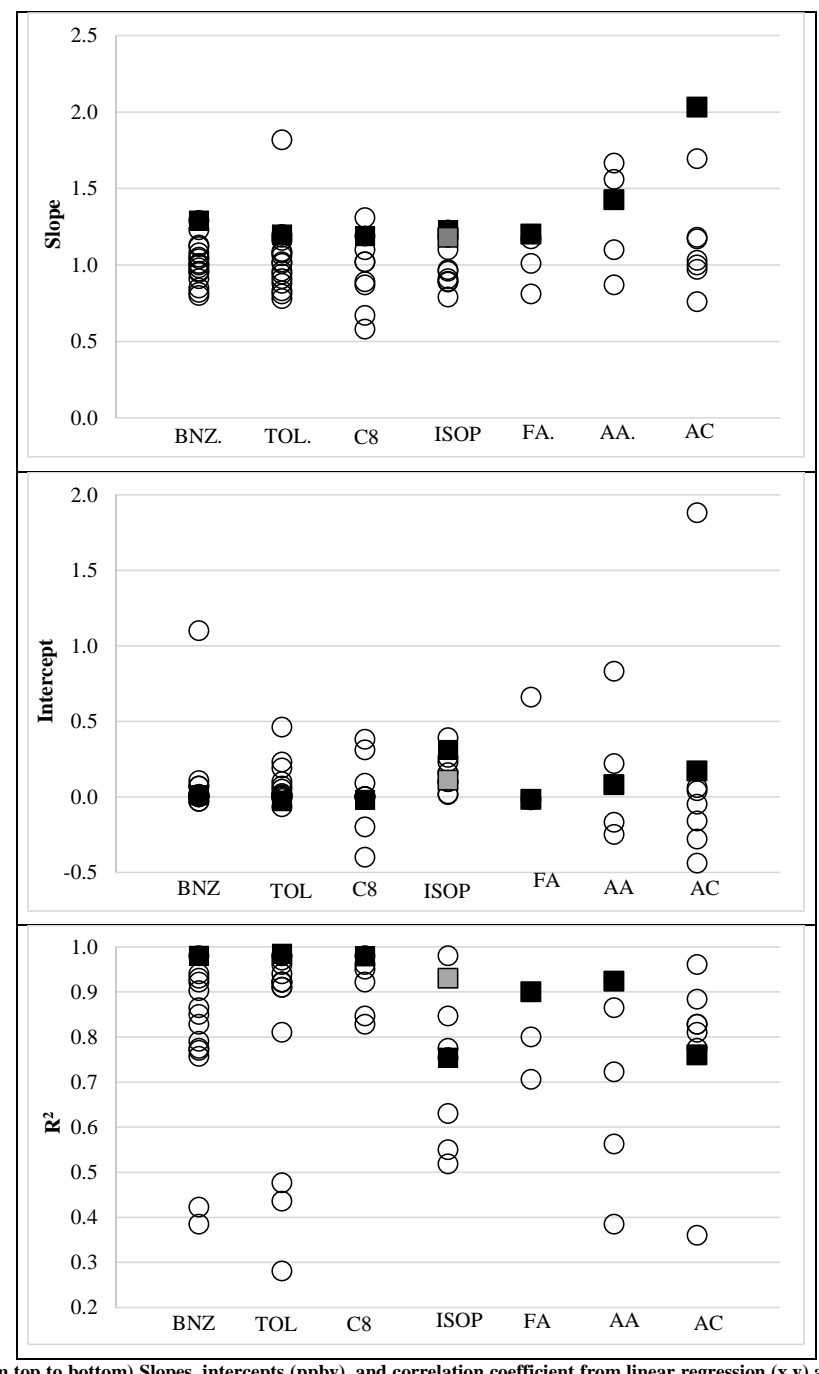

**Figure 2. (From top to bottom) Slopes, intercepts (ppbv), and correlation coefficient from linear regression (x,y) analyses between PTR-MS (y) and independent VOC measurement techniques (x), from this study (black squares), and other published studies (open**
5 **circles). BNZ – benzene; TOL – toluene; C8 – C8 aromatics; ISOP – isoprene; FA – formaldehyde; AA – acetaldehyde; AC – acetone. Note: grey squares are determined from analysis of isoprene afternoon data from this study. Published studies used in this figure are referred within text for each compound subsection (3.111 – 3.2.3).**



**Figure 3.** Time series of formaldehyde, acetaldehyde acetone measured in the DNPH-HPLC samples in SPS2. Bottom panel: time series of the extraction masss (g) of the DNPH cartridge samples.



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
