# Peer review of "Comparison of VOC measurements made by PTR-MS, Adsorbent Tube/GC-FID-MS and DNPH-derivatization/HPLC during the Sydney Particle Study, 2012: a contribution to the assessment of uncertainty in current atmospheric VOC measurements"

_Atmospheric Measurement Techniques, 2016_

## Referee Comment (RC1) · Anonymous Referee #2 · 24 Feb 2017

General comment: This manuscript presents data from a comparison of VOC measurements made by PTR-Q-MS, Adsorbent tube GC-FID/MS and DNPH-HPLC in an urban area. The VOC selected for comparison were C6- to C8-aromatics, isoprene and C1- to C3-carbonyl compounds. Compared to the two other methods the PTR-MS measurements were found to overestimate the VOC mixing ratios by a factor of 1.18 to 2.01. Most of the discussion is based on the assumption of positively biased

[Figure]

PTR-MS measurements due to possible isobaric compounds or fragments on the respective m/z signals. Therefore the authors mainly apply data corrections considering these interferences to reduce the observed overestimation by PTR-MS. Literature data on observed interferences in PTR-MS measurements combined with potentially interfering compounds measured by AT-GC-FID/MS were taken into account to correct the PTR-MS mixing ratios. This procedure improved the comparison. A critical review of the AT-GC_FID/MS measurement is not taken into account. The comparison of the PTR-MS data to the DNPH-HPLC data discusses two possible reasons for the observed regression slope of > 1. An overestimation of PTR-MS measurements due to possible isobaric compounds or fragments, or an underestimation of the DNPH-HPLC measurements. It is stated but not proven that the respective m/z signals of the PTR-MS measurements were dominated by the carbonyl compound of interest and on the other hand the underestimation by the DNPH-method cannot be ruled out. In the case of DNPH-acetone measurements, there are indications but no proof for a negative bias due to high humidities.

Specific comments: Chapter 3.1 Inter-comparison of PTR-MS and AT-VOC samples analysed by GC-FID-MS: The discussion is solely based on the assumption that PTR-MS is biased by positive interferences to explain the results. Can underestimation by the AT-VOC technique be ruled out? If so please discuss. Page 7 Line 24: The authors use the fragmentation patterns for alkylbenzenes to correct their data adapted from Gueneron et al. (2015) which provides those fragmentation patterns for E/N 80 Td and E/N 120 Td. While the PTR used has an E/N of ∼100 Td (V(Drift) = 445 V, T(Drift) = 60°C, p(Drift) = 2.16 mbar) the literature values for E/N 120 Td were used which imply a high fragmentation and therefor overestimates the contribution of m/z 79 from alkylbenzenes. Therefore the slope of 1.27 is only a lower estimate. Figure 3: The extraction mass of the DNPH cartridge sample is estimated to be increased by condensed water. Please plot the dew point / rH with the data. As the DNPH cartridges were sampled at 7°C the extraction mass should correlate with the dew point. All datasets exceeding a dew point of 7°C should be omitted or lab tests of the

influence of condensing water on the sampling/derivatization should be included in the discussion. Page 12 Line 12: Please proof the statement 'Overall, the PTR-MS signal at m/z 59 was dominated by acetone'

Technical comments: Page 1 Line 17 and Page 12 Line 23: Non consistent values: median of 0.13 vs. median of 0.11 Page 1 Line 20: A slope with a value of 1.08 stated as the lower range of slopes is not stated in the following text/tables. Page 4 Line 10: 'Cheng et al. (2008a)' is missing in the reference list. Page 4 Line 15: Remove 'and' from the sentence 'a PAMS gas standard (Spectra Gases, Linde NJ USA) and were used' Page 4 Line 25: 'Ellis and Mayhew 2014' is missing in the reference list. Table 2: The uncertainty of the calibration factors is 6% or higher. This is not reflected in the number of significant digits of the given values Page 6 Line 10: Add 'of' to the sentence 'The results this inter-comparison' Page 6 Line 11: Add 'are' to the sentence 'conclusions about the uncertainty in current VOC measurements presented' Page 7 Line 3: 'benzene data yielded a slope of $1.47 \pm 0.04$'; Figure 1a and table 5 provides a slope off 1.27. Which number is correct? If Figure 1a and table 5 provides an already corrected dataset please state so in the captions. Page 7 Line 7: 'Slope of $\sim$1.5' → see previous comment Page 7 Line 13: Rephrase sentence Page 9 Line 29: Remove 'Ne' at the end of the line Page 9 Line 38: Change 'din' to 'in' Page 12 Line 15: Please complete the sentence: 'guarded against as stated' Page 12 Line 20: The statement 'Inter-comparisons have been made between three independent techniques' is not correct. Although 3 different techniques were used the inter-comparison took only place between two techniques for each of the evaluated compounds. Please clarify.

---

## Referee Comment (RC2) · Anonymous Referee #3 · 3 Mar 2017

This paper describes an inter-comparison of PTR-MS, Adsorbent Tube analysis by GC-FID and DNPH-HPLC methods in an urban environment in Sidney Australia. Inter-comparisons between different techniques are always an important exercise and should be done whenever possible to identify potential issues that affect the data interpretation. The methods compared here are the basic standard methods; the PTR-MS is the older quadrupole version and not the newer PTR-TOF, the AT analysis is done

with a GC-FID and not with a more sophisticated GC such as a GCxGC-MS, but this means that they are widely used and available to the community. This inter-comparison reveals some known artifacts from the DNPH method and discusses some potential interferences for the PTR-MS method, although I do not fully agree with the conclusion by the authors that interfering compounds are the main reason for the general higher means or slopes for the PTR-MS compared to the other techniques. There are also a few other points or omissions that need to be addressed, before this manuscript can be accepted. I actually think some minor additional measurement tests would be appropriate to resolve some of the open issues for the disagreement between the methods. Overall a major revision to the paper is needed.

Major comments

The authors conclude that interferences from other mainly unknown compounds are the main reason for higher PTR-MS measurements compared to the other techniques, even though they take all the known interferences into account. The amount of work that has been done on PTR-MS and recently on PTR-TOF-MS, including work on measurements of coupled GCs with PTR-MS, give us a very good understanding of the compounds that can interfere, especially for benzene, toluene, the C8-aromatics and acetaldehyde. All this published work has not revealed any interferences that can make up an additional ∼20% bias in PTR-MS in addition to what has been taken into account in this work already. I suggest revising the conclusion and various places in the text accordingly or show proof or possibilities for other interferences. One possibility for benzene would be the acetic acid cluster (propionic acid for toluene), which could be significant under the configuration of the PTR-MS used here. The E/N seemed to have been around 100 Td, which is pretty low and could favor cluster ions. These potential interferences need to be also added in the discussion.

It is also clear that in the presented literature summary of intercomparisons presented in Figure 2, the current study is always at the high end. This also could be an indication for a more systematic overestimation of PTR-MS or underestimation of the AT and

DNPH methods. I would like to see that pointed out in the text and conclusion more clearly. Here I should mention that I like Figure 2 a lot and would like to see this expanded a little by adding a Table with all the data and references included.

Formaldehyde and acetone do seem to have significant interferences in PTR-MS, some of which have not been discussed in the text well. Stoenner et al 2016 (DOI 10.1002/jms.3893) have described glyoxal measurements by PTR-TOF-MS and found that glyoxal is detected mostly at mass 31, which could give a significant interference for formaldehyde, and only to a small fraction on mass 59. In addition, the glyoxal reaction is not exothermic as stated in the text, just suffers from strong fragmentation. There are several places in the manuscript that have to be updated with the results from this paper.

The most obvious interference is seen for isoprene. In an urban environment or oil&gas extraction regions the largest interference likely comes from cycloalkanes, which are a significant fraction in vehicle exhaust. The detection of cycloalkanes on mass 69 was described first by Yuan et al 2014 (http://dx.doi.org/10.1016/j.ijms.2013.11.006). This has been overlooked in the discussion and needs to be added.

The correction of benzene and toluene has been done using literature data, but nothing has been done to confirm the interferences or fragmentations in the PTR-MS used here. The instrument settings influencing the E/N are critical for the fragmentation of larger aromatics to benzene and monoterpenes to toluene. The E/N used here is very low and fragmentation should be reduced compared to the literature data used. Most importantly, the fragmentation in the PTR-MS at a certain setting can be measured very easily by simply exposing the instrument to the pure compound. Using the measured fragmentation patterns and looking at the parent masses 121 or 137 measured during SPS2 should give a much better indication on potential contributions to benzene and toluene and can be used to properly correct mass 79 and 93 in PTR-MS.

If the PTR-MS measurements suffer from interferences, the PTR-MS should be cor-

rected and not the AT-VOC data. You are generating not atmospherically meaningful data. Please subtract from the PTR-MS benzene and toluene signal instead. Also, it needs to be made clear in the figures and tables that corrected data were used. This means changes the formula. Text, table and figures in many places.

The same issues arise in the discussion about DNPH losses. If for example acetone is collected poorly on DNPH, especially at high water mixing ratios, a very simple test with an acetone standard should be done. It seems from the instrument description that only the HPLC was calibrated with a liquid standard, but a calibration should be done with the entire sampling setup. This means including the DNPH cartridges. You actually should cross calibrate the PTR-MS standard. From page 5 line 10 it seems that cross calibration was done only with the BTEX standard and the GC-FID PTR-MS combination and also not through the adsorbent tubes.

In addition, the DNPH measurements of acetone in Figure 3 actually go to essentially zero at high humidity. In the atmosphere, especially in urban areas, so low acetone values have never been observed and clearly point to measurement issues. This should be added as an argument for the ketone loss in the DNPH method.

The selection criteria for measurement comparison #4 of median/MDL>5 should not be used. All the data that are above the MDL should be used. That's what MDL means. If you do have a reason not to use those data, you have to change your definition of MDL. The use of these lower concentration data should decrease the R-coef, but the slopes should still be valid.

Minor comments

Page 4 line 21: Was the silcosteel inlet tube heated? If not, this is highly recommended.

Page 4 line25: Ellis and Mayhew 2014 ref is missing from the reference list, Lindinger et al 1998 or de Gouw and Warneke 2007 could be mentioned here as well.

Page 4 line 41: How does your definition of MDL compare to the more standard S/N=2

or 3?

Page 5 line4: Table 2 comes before Table 1 in the text, they should be switched.

Page 5 line 4: What do you mean with the scatter in the calibration measurements? Was this 10% around the mean for the 30 min daily measurements or are the 10% the deviation for each daily calibration around the campaign average?

Page 5 line 10: Can you add the manufacturers stated uncertainties for the calibration standards?

Table 2: The sensitivity of PTR-MS should not be given in ncps/ppb only without indicating the primary ion counts or adding cps/ppb. With ncps/ppb you still don't know the actual instrument sensitivity.

Page 5 line 18: "Each compound known to substantially contribute" How do you know which compounds contribute, I guess from literature studies? Please add.

Page 5 line 3: What is the difference between RMA and ODR, which is mostly used for instrument comparisons? In ODR the actual instrument uncertainty can be used as the error estimate for both axis. Is that also done in RMA?

Table 3: Please indicate the sampling times and PTR-MS averaging times.

Page 7 line 24: Here and elsewhere in the text: Cappellin et al 2010 is the wrong reference. This should be Cappellin 2012.

Page 8 line 33: "we correct the AT-VOC data". This should say something like: We calculate the weighted sum of the individual VOCs measured by AT-VOC that corresponds to m/z 107

Page 9 line 15: If you separate daytime data, you should also look at nighttime data, at night the interference should be the largest.

Page 9 line 33: I don't think the influence of other compounds on isoprene by PTR-MS

[Figure]

is poorly understood, it is indeed a poorly quantified uncertainty, but we know pretty well what the other compounds are.

Figure 2: Indicate what the grey squares are. I guess daytime data.
* * *

---

## Referee Comment (RC3) · Anonymous Referee #1 · 7 Mar 2017

General Comments

In this paper the authors present results of an intercomparison of three methods for analyzing volatile organic compounds, including PTR-MS, adsorption tube-GCFID, and DNPH-HPLC. Data were obtained from a 2012 field study in Sydney, Australia. The degree of agreement between methods was evaluated based on comparison of slopes

and intercepts of plots of one method against another. In some cases agreement was within 95% confidence limits, and in others not. Discrepancies were typically explained as being due to contributions of non-target compounds to the ions used to quantify compounds by PTR-MS (high bias) or loss of compounds during DNPH cartridge sampling (low bias). Overall, the measurements were carefully done, systematic, and the comparison was statistically sound and thorough. The explanations for discrepancies were reasonable and in many cases supported by observations reported by others for these methods. The manuscript is concise and well written, and I think makes a useful contribution to the literature on atmospheric measurement methodology. It is essentially publishable in AMT in its current form, though I note a few typos below.

Specific Comments

None.

Technical Comments

1. Page 9, line 29: Delete "Ne" at end of sentence.

2. Page 9, line 38: "din" should be "in".

3. Page 11, line 27: I think "AT-VOC" should be " DNPH-HPLC".

---

## Author Comment (AC1) · 2 Oct 2017

Authors' responses to all referee comments are on a single file uploaded as supplementary file.

[Figure]

Please also note the supplement to this comment:
https://www.atmos-meas-tech-discuss.net/amt-2016-349/amt-2016-349-AC1-supplement.pdf

————————————————————

---

## Author Comment (AC2) · 2 Oct 2017

**Comparison of VOC measurements made by PTR-MS, Adsorbent Tube/GC-FID-MS and DNPH-derivatization/HPLC during the Sydney Particle Study, 2012: a contribution to the assessment of uncertainty in current atmospheric VOC measurements**

**Authors Response to review comments**

Erin Dunne[1], Ian E. Galbally[1], Min Cheng[1], Paul Selleck[1], Suzie B. Molloy[1], Sarah J. Lawson[1]

[1]CSIRO Oceans and Atmosphere, Aspendale, 3195, Australia

*Correspondence to*: Erin Dunne (erin.dunne@csiro.au)

The authors would like to thank the three referees for their thoughtful feedback. The review process has been of great benefit to the current manuscript as well as led to improvements in our practice of conducting atmospheric measurements of VOCs which has flowed into current measurement programs. The referee comments (shown in bold) are addressed in turn below.

**Anonymous Referee #1**

**General Comments**

**In this paper the authors present results of an intercomparison of three methods for analysing volatile organic compounds, including PTR-MS, adsorption tube-GCFID, and DNPH-HPLC. Data were obtained from a 2012 field study in Sydney, Australia. The degree of agreement between methods was evaluated based on comparison of slopes and intercepts of plots of one method against another. In some cases agreement was within 95% confidence limits, and in others not. Discrepancies were typically explained as being due to contributions of non-target compounds to the ions used to quantify compounds by PTR-MS (high bias) or loss of compounds during DNPH cartridge sampling (low bias). Overall, the measurements were carefully done, systematic, and the comparison was statistically sound and thorough. The explanations for discrepancies were reasonable and in many cases supported by observations reported by others for these methods. The manuscript is concise and well written, and I think makes a useful contribution to the literature on atmospheric measurement methodology. It is essentially publishable in AMT in its current form, though I note a few typos below.**

**Specific Comments**
**None.**

**Technical Comments**
1. **Page 9, line 29: Delete "Ne" at end of sentence.**
   Corrected
2. **Page 9, line 38: "din" should be "in".**

Corrected

3. **Page 11, line 27: I think "AT-VOC" should be " DNPH-HPLC".**
   Corrected

**Anonymous Referee # 2**

**General comment:**

**This manuscript presents data from a comparison of VOC measurements made by PTR-Q-MS, Adsorbent tube GC-FID/MS and DNPH-HPLC in an urban area. The VOC selected for comparison were C6- to C8-aromatics, isoprene and C1- to C3 carbonyl compounds. Compared to the two other methods the PTRMS measurements were found to overestimate the VOC mixing ratios by a factor of 1.18 to 2.01. Most of the discussion is based on the assumption of positively biased PTR-MS measurements due to possible isobaric compounds or fragments on the respective m/z signals. Therefore the authors mainly apply data corrections considering these interferences to reduce the observed overestimation by PTR-MS. Literature data on observed interferences in PTR-MS measurements combined with potentially interfering compounds measured by AT-GC-FID/MS were taken into account to correct the PTR-MS mixing ratios. This procedure improved the comparison. A critical review of the AT-GC_FID/MS measurement is not taken into account.**

There were two methods of determining uncertainties in VOC measurements assessed in this study, bottom-up and top-down. The bottom-up uncertainty analysis proceeded via the mathematical model as described in the Guide to Expression of Uncertainty in Measurement (JCGM, 2008). In the second approach to assessing uncertainty, the top-down method, we evaluated the systematic difference between two methods by evaluating the slope and intercept of a linear regression between two sets of paired simultaneous measurements.

Contributions to the uncertainty of these measurements that were not included in the bottom-up analyses but were apparent from the top-down analyses are discussed. Some examples of these for PTR-MS and DNPH are identified. None were immediately apparent for AT-VOC.

When comparing PTR-MS measurements to more selective VOC measurement techniques such as chromatographic methods, the presence of interference from non-target compounds in the target ion mass signal often results in an apparent positive bias in the PTR-MS reported values. The uncertainty due to interference was not incorporated into the bottom-up uncertainty analysis but was investigated here as a possible reason for the discrepancy between PTR-MS and the other two measurement systems.

Likewise, possible interference due to the presence of liquid water in the DNPH measurement system was not included in the bottom-up uncertainty assessment and its role in causing the apparent discrepancy between the PTR-MS and DNPH measurements of carbonyls was discussed.

*The uncertainty due to the apparent bias in the PTR-MS and DNPH techniques discussed in this study were identified as uncertainty that is poorly understood and poorly quantified by the bottom-up uncertainty analysis method. The text will be revised to communicate this more clearly.*

In response to the referee comment "A critical review of the AT-GC_FID/MS measurement is not taken into account", the authors agree more detail is required for the AT-VOC methodology to justify the statement that no apparent bias was identified for the AT-VOC method.

The bottom-up uncertainty analysis for the AT-VOC method included uncertainty due to uncertainty in:

- the accuracy of the certified calibration standards
- variance in the response factors of the GC-FID in measurements of certified calibration gas standards

- uncertainty in the loop volume, temperature and pressure
- variance in a series of replicate ambient measurements of the target VOCs by the AT-VOC method

The revised manuscript will include additional detail in the AT-VOC methodology related to: sample flow rate and safe sample volumes; flow calibrations and sample volume; treatment of field blanks; sample storage; and desorption efficiency. These details will be added to the methodology section 2.3 to provide more confidence in the AT-VOC method employed in this study. For instance:

Tubes were conditioned and pre-analysed prior to use. Two tubes in series were installed for every sample to check breakthrough for each analyte was <5% for all samples. One field blank and one lab blank per ten samples were collected during the sample period. In order to capture potential contamination during transport, storage and handling field blank tubes were uncapped and installed in the automated sampler for the same period as the samples. No flow was passed through the blank tubes during the deployment period. Prior to and following sampling tubes were capped and stored in an air-tight metal tins at < 4°C.]

The automated sampler and the PTR-MS sampled air via silco steel inlets of almost equal length both connected to a common glass inlet manifold. The gravimetrically prepared Apel Reimer standard used to calibrate the PTR-MS, was also analysed with the GC-FID-MS against a certified BTEX gas. The FID response factors for the 2 standards differed by 5 – 9% (BTEX/Apel Reimer Ratios: benzene 0.95; toluene 0.95 and m-xylene 0.91) and we can conclude that the PTR-MS and GC-FID-MS calibrations were compatible within these limits. Consequently, systematic differences due to inlet performance or instrument calibrations were not expected to be major factors in the discrepancy between the two methods.

The difference between the PTR-MS and AT-VOC reported values for BTEX did not correlate with any meteorological parameters measured including relative humidity, temperature, wind speed, wind direction.

**The comparison of the PTR-MS data to the DNPH-HPLC data discusses two possible reasons for the observed regression slope of > 1. An overestimation of PTR-MS measurements due to possible isobaric compounds or fragments, or an underestimation of the DNPH-HPLC measurements. It is stated but not proven that the respective m/z signals of the PTRMS measurements were dominated by the carbonyl compound of interest and on the other hand the underestimation by the DNPH-method cannot be ruled out. In the case of DNPH-acetone measurements, there are indications but no proof for a negative bias due to high humidity.**

In response to the referee comment "It is stated but not proven that the respective m/z signals of the PTRMS measurements were dominated by the carbonyl compound of interest and on the other hand the underestimation by the DNPH-method cannot be ruled out."

The m/z signals of the PTR-MS were considered to be dominated by the compound of interest according to the following criteria:

- The compound of interest is known to produce an ion signal at the target m/z in PTR-MS
- A priori knowledge of other VOCs commonly measured in the atmosphere which may be detected at the target m/z in PTR-MS.
- High $R^2$ values in comparison with more selective chromatographic techniques:
  - The high $R^2$ values of 0.92 give confidence that both the PTR-MS and the DNPH technique were both responding to formaldehyde and acetaldehyde.
  - Acetone samples that were affected by condensation have been removed from the dataset by removing samples obtained when dewpoint was > the automated sampler chiller temp of 7°C used in this study (see comment below). Removing this data resulted in an $R^2$ of 0.89 which gives us confidence that the PTR-MS signal at m/z 59 is dominated by acetone.

A response to the referee **comment "In the case of DNPH-acetone measurements, there are indications but no proof for a negative bias due to high humidity "** is provided in the responses to specific comments below.

5 **Specific comments:**

**Chapter 3.1 Inter-comparison of PTR-MS and AT-VOC samples analysed by GC-FID-MS: The discussion is solely based on the assumption that PTRMS is biased by positive interferences to explain the results. Can underestimation by the AT-VOC technique be ruled out? If so please discuss.**

See response to major comments above.

**Page 7 Line 24: The authors use the fragmentation patterns for alkylbenzenes to correct their data adapted from Gueneron et al. (2015) which provides those fragmentation patterns for E/N 80 Td and E/N 120 Td. While the PTR used has an E/N of ~100 Td (V(Drift) = 445 V, T(Drift) = 60 C, p(Drift) = 2.16 mbar) the literature values for E/N 120 Td were used which imply a high fragmentation and therefor overestimates the contribution of m/z 79 from**
15 **alkylbenzenes. Therefore the slope of 1.27 is only a lower estimate.**

This is a valid criticism and in lieu of better data the authors propose using an estimated value based on the average of the branching ratios at 80Td and 120 Td observed by Gueneron et al (2015).

**Figure 3: The extraction mass of the DNPH cartridge sample is estimated to be increased by condensed water. Please**
20 **plot the dew point / rH with the data. As the DNPH cartridges were sampled at 7 C the extraction mass should correlate with the dew point. All datasets exceeding a dew point of 7C should be omitted or lab tests of the influence of condensing water on the sampling/derivatization should be included in the discussion.**

The referee also stated **"In the case of DNPH-acetone measurements, there are indications but no proof for a negative bias due to high humidity."**

25

Section 3.2. provided evidence of the relationship between extraction mass and low DNPH reported acetone concentrations. The text states that the additional extraction mass was due to condensation forming in the chilled DNPH cartridges. In section 3.2.3 the authors also provide a mechanism for the acetone loss process via back reaction of the hydrazone derivative intermediate with water. However, as the referee states this provides an indication but not proof for a negative bias in DNPH
30 reported acetone concentrations due to condensation The following provides an expanded discussion of the identified negative bias incorporating suggestion from the referees which will be included in the revised manuscript.

Typically the mass of the DNPH cartridge extraction is ~2.0 g, at high relative humidity the mass of the extraction from the DNPH cartridge was observed to be higher (~2.1 – 2.4 g) (see top panel of figure below). Higher extraction mass corresponded
35 with DNPH acetone concentrations of close to zero which would be extremely unlikely in an urban area and point to a significant measurement issue.

The compartment housing the DNPH cartridges in the automated sampler was maintained at ~7° C and the cartridges were refrigerated before and after sampling. Liquid water was observed in the affected cartridges on retrieval and it was assumed the additional mass was due to the condensation of water from ambient air in the chilled DNPH cartridge which was more
40 pronounced at high RH. The higher extraction mass was correlated with the dewpoint temperature with higher extraction masses occurring at dewpoint temp > 7C (Panel B), the temperature at which the DNPH cartridges were sampled. DNPH reported concentrations of acetone frequently approached zero at dewpoint temperatures > 15C, and the discrepancy between PTR-MS and DNPH reported values for acetone increased with increasing dewpoint (Panel C).

As suggested by anonymous referee 2, data points with average dewpoint temperatures > 7C were omitted from the RMA analysis. The average dewpoint temperature was < 7C in only 11 out of 53 DNPH samples, resulting in a significantly reduced datatset. However, omitting datapoints with dew point > 7 C markedly improved the agreement between the DNPH and PTR-MS measurements of Acetone (Panels C and D below), with the results of the RMA analysis changing from Slope = 2.01, R2 = 0.76 (N=53), to slope = 1.4, R2 = 0.89 (N = 11) when water affected samples were omitted.

Clearly significant measurement issues with acetone remain which require further investigation, however the author's believe this analysis provides proof of a negative bias in the DNPH acetone measurements due to reduction of the collection efficiency of acetone on DNPH in the presence of liquid water.

[Figure]

[Figure]

**Page 12 Line 12: Please proof the statement 'Overall, the PTR-MS signal at m/z 59 was dominated by acetone'**

See response to Referee #2 Major Comments above.

**Technical comments:**

**Page 1 Line 17 and Page 12 Line 23: Non consistent values: median of 0.13 vs. median of 0.11**

Corrected- the correct value is 0.11 ppbv

**Page 1 Line 20: A slope with a value of 1.08 stated as the lower range of slopes is not stated in the following text/tables.**

The correct range of the slopes is 1.16 – 2.01, and text will be updated accordingly.

**Page 4 Line 10: 'Cheng et al. (2008a)' is missing in the reference list.**

This reference was incorrectly dated. Should be Cheng et al (2016) (doi:10.1111/ina.12201). However on review this reference does not provide any substantial detail to the method description provided in the present manuscript.

**Page 4 Line 15: Remove 'and' from the sentence 'a PAMS gas standard (Spectra Gases, Linde NJ USA) and were used'**

Corrected

**Page 4 Line 25: 'Ellis and Mayhew 2014' is missing in the reference list. Table 2:**

Corrected

**The uncertainty of the calibration factors is 6% or higher. This is not reflected in the number of significant digits of the given values**

The scatter is the relative std deviation mean for the all calibrations across the campaign. Table 2 will be revised to present this in absolute terms as ncps instead of a percentage.

**Page 6 Line 10: Add 'of' to the sentence 'The results this inter-comparison'**

Corrected

**Page 6 Line 11: Add 'are' to the sentence 'conclusions about the uncertainty in current VOC measurements presented'**

Corrected

**Page 7 Line 3: 'benzene data yielded a slope of 1.47  0.04'; Figure 1a and table 5 provides a slope off 1.27. Which number is correct? If Figure 1a and table 5 provides an already corrected dataset please state so in the captions.**

The uncorrected data yielded a slope of $1.47 \pm 0.04$, and the corrected data (recently revised) yielded a slope of $1.31 \pm 0.03$. The Tables and figures will be corrected and properly named accordingly.

**Page 7 Line 7: 'Slope of 1.5' ! see previous comment**

Corrected

**Page 7 Line 13: Rephrase sentence**

Deleted: In a separate study, the PTR-MS was exposed to a certified gas standard containing roughly equivalent VMRs of benzene, ethylbenzene, propyl- and isopropyl-benzene, among other components. The signal at m/z 79 was 41% higher than in measurements of a standard containing benzene but not ethylbenzene, propyl- and isopropyl-benzene also tested.

This phrase does not add anything quantitative to the discussion and was deleted.

**Page 9 Line 29: Remove 'Ne' at the end of the line**

Corrected

**Page 9 Line 38: Change 'din' to 'in'**

Corrected

**Page 12 Line 15: Please complete the sentence: 'guarded against as stated'**

No change, the text reads:

"At high humidity, the formation of condensation in DNPH cartridges must be guarded against as stated in TO-11A (USEPA, 1999)."

**12 Line 20: The statement 'Inter-comparisons have been made between three independent techniques' is not correct. Although 3 different techniques were used the inter-comparison took only place between two techniques for each of the evaluated compounds. Please clarify.**

Deleted: Inter-comparisons have been made between three independent techniques covering the measurement in the atmosphere of benzene, toluene, $C_8$ aromatics, isoprene, formaldehyde, acetaldehyde and acetone.

Inserted: Comparisons have been made between measurements of benzene, toluene, $C_8$ aromatics, and isoprene by two independent techniques: PTR-MS and adsorbent tube sampling with GC-FID-MS analysis. Also, PTR-MS measurements of formaldehyde and acetone were compared with a DNPH derivatization method using HPLC analysis.

**Anonymous Referee #3**

**General Comments:**

**This paper describes an inter-comparison of PTR-MS, Adsorbent Tube analysis by GC-FID and DNPH-HPLC methods in an urban environment in Sydney Australia. Inter-comparisons between different techniques are always an important exercise and should be done whenever possible to identify potential issues that affect the data interpretation. The methods compared here are the basic standard methods; the PTR-MS is the older quadrupole version and not the newer PTR-TOF, the AT analysis is donewith a GC-FID and not with a more sophisticated GC such as a GCxGC-MS, but this means that they are widely used and available to the community. This inter-comparison reveals some known artifacts from the DNPH method and discusses some potential interferences for the PTR-MS method, although I do not fully agree with the conclusion by the authors that interfering compounds are the main reason for the general higher means or slopes for the PTR-MS compared to the other techniques. There are also a few other points or omissions that need to be addressed, before this manuscript can be accepted. I actually think some minor additional measurement tests would be appropriate to resolve some of the open issues for the disagreement between the methods. Overall a major revision to the paper is needed.**

**Major comments**

**The authors conclude that interferences from other mainly unknown compounds are the main reason for higher PTR-MS measurements compared to the other techniques, even though they take all the known interferences into account. The amount of work that has been done on PTR-MS and recently on PTR-TOF-MS, including work on measurements of coupled GCs with PTR-MS, give us a very good understanding of the compounds that can interfere, especially for benzene, toluene, the C8-aromatics and acetaldehyde. All this published work has not revealed any interferences that can make up an additional 20% bias in PTR-MS in addition to what has been taken into account in this work already. I suggest revising the conclusion and various places in the text accordingly or show proof or possibilities for other interferences.**

The authors agree that interference by unknown non-target compounds cannot fully explain the remaining discrepancy between the measurement PTR-MS and the DNPH/AT-VOC methods. The reasons for the remaining discrepancy between the measurement systems remains unknown.

As pointed out by the referee in general comments above some **"minor additional measurement tests would be appropriate to resolve some of the open issues for the disagreement between the methods".**

The authors agree further tests would have been very beneficial in resolving these issues however such tests were not conducted at the time of the study in 2012. As a result of the process undertaken for writing this manuscript improved testing procedures have been incorporated into present measurement campaigns but the data is not currently available for use in this manuscript. A list of technical suggestions will be included in the conclusions of the revised manuscript outlining recommendations arising
5    from this study.

**One possibility for benzene would be the acetic acid cluster (propionic acid for toluene), which could be significant under the configuration of the PTR-MS used here. The E/N seemed to have been around 100 Td, which is pretty low and could favour cluster ions. These potential interferences need to be also added in the discussion.**
10   The contribution of the acetic acid cluster ion is expected to be minor (~1% of Acetic Acid ion signal) at the operating conditions used in this study (100Td) (Haase et al., 2012).

**It is also clear that in the presented literature summary of intercomparisons presented in Figure 2, the current study is always at the high end. This also could be an indication for a more systematic overestimation of PTR-MS or**
15   **underestimation of the AT and DNPH methods. I would like to see that pointed out in the text and conclusion more clearly.**
The discussion and conclusions will be adjusted to reflect that "this analysis identified some sources of uncertainty that were poorly understood and poorly quantified in the bottom-up measurement uncertainty analysis. These sources of uncertainty include the contributions from non-target compounds to the measurement of target compounds in PTR-MS; and, the under-
20   reporting of formaldehyde acetaldehyde and acetone by the DNPH technique. As well as these, this study has identified a specific interference by liquid water in the measurement of acetone by DNPH method. However, even when the bias due to these 3 factors was taken into account the PTR-MS measurements were systematically ~20% higher than the AT-VOC and DNPH measurements indicating substantial measurement uncertainty remains which is unresolved."
**Here I should mention that I like Figure 2 a lot and would like to see this expanded a little by adding a Table with all**
25   **the data and references included.**
The referee is referred to Cui et al (2016, Atmos. Meas. Tech. 9, 5763 – 5799) which provides a table with a majority of this information included with the exception of data for isoprene. We feel such a Table would be duplication however we would consider including it if required.

30   **Formaldehyde and acetone do seem to have significant interferences in PTR-MS, some of which have not been discussed in the text well. Stoenner et al 2016 (DOI 10.1002/jms.3893) have described glyoxal measurements by PTR-TOF-MS and found that glyoxal is detected mostly at mass 31, which could give a significant interference for formaldehyde, and only to a small fraction on mass 59. In addition, the glyoxal reaction is not exothermic as stated in the text, just suffers from strong fragmentation. There are several places in the manuscript that have to be updated with the results from**
35   **this paper.**
We are grateful for the referee bringing this work to our attention. The text in the formaldehyde discussion will be updated as follows:
In PTR-MS formaldehyde is detected at m/z 31. The measurement of formaldehyde with PTR-MS is complex as its proton transfer chemical ionization reaction with $H_3O^+$ is close to endothermic and loss via back reaction in humid air is non-negligible
40   (Hansel et al., 1997; Inomata et al., 2008). In order to account for the water vapour dependence of the PTR-MS response to formaldehyde daily instrument background and calibration measurements were made using zero air that had the same mole fractions of $H_2O$ as the ambient air being sampled. The linear relationship observed between the formaldehyde calibration

factors measured daily and the respective water vapour density (g m$^{-3}$) was determined, and a corrected calibration factor was applied to the ambient hourly data based on the ambient water vapour density measured hourly.

RMA regression analysis between the PTR-MS signal at m/z 31 and the formaldehyde in the DNPH-HPLC samples yielded a slope of 1.30 ± 0.04, intercept = -0.07 ± 0.01 ppbv, $R^2$ = 0.92, and RMSD = 0.14 ppbv (N = 77). The high $R^2$ value gives confidence that both the PTR-MS and the DNPH technique were both responding to formaldehyde. The comparisons presented in Table 4 indicate that the mean values reported by each instrument agree within 95% confidence limits.

To examine any possible effect of liquid water, the analysis was repeated excluding the data 16/4 to 24/4. The results yielded a slope of 1.20 ± 0.04, intercept = -0.02 ± 0.02 ppbv, $R^2$ = 0.90, and RMSD = 0.16 ppbv (N = 53) (Table 5 and Figure 1f) indicating a minor but significant effect of liquid water.

The slope of 1.2 may be a result of contributions to the PTR-MS signal at m/z 31 from compounds other than formaldehyde. Inomata et al (2008) described a procedure to correct the m/z 31 ion signal for contributions of methanol, ethanol, and methyl hydroperoxide which are known to produce fragment ions at m/z 31 in PTR-MS. The dominant ion signal in the PTR-MS spectra of glyoxal is detected at m/z 31 due to strong fragmentation of the parent ion (Stoenner 2017). However, these authors also found that like formaldehyde, glyoxal also has a low proton affinity loss via back reaction in humid air is also non-negligible resulting in very low PTR-MS sensitivity of ~0.3 – 0.8 ncps/ppbv compared to a formaldehyde sentivity of ~1.4 ncps/ppbv for this study. The concentration of glyoxal measured in the DNPH samples was on average ~20% of the corresponding formaldehyde concentration in this study and given it's low PTR-MS response it is likely to make a negligible contribution to the signal at m/z 31 observed.

Applying the same correction procedure described by Inomata to the reduced data set (N= 53) in this study from Sydney 2012 significantly improved the slope (slope = 1.00) but resulted in a large negative off set of -0.15 ± 0.03 and poorer correlation ($R^2$ = 0.83).

Previous studies have reported PTR-MS values for formaldehyde that were systematically higher than DNPH-HPLC measurements (Wisthaler et al. 2008, Cui et al. 2016) and higher than DOAS and Hanzstch techniques (Wisthaler et al. 2008, Warneke et al 2011). Other studies report DNPH-HPLC values for formaldehyde that were systematically lower than those reported by other analytical methods (DOAS, FTIR, Hantzsch, TDLAS)(Kleindienst et al., 1988; Lawson et al., 1990; Gilpin et al., 1997; Hak et al., 2005; Wisthaler et al., 2006).

In summary, a comparison between the measurements of formaldehyde by PTR-MS and the DNPH technique indicates there was not a significant difference in the measured concentrations.

**The most obvious interference is seen for isoprene. In an urban environment or oil&gas extraction regions the largest interference likely comes from cycloalkanes, which are a significant fraction in vehicle exhaust. The detection of cycloalkanes on mass 69 was described first by Yuan et al 2014 (http://dx.doi.org/10.1016/j.ijms.2013.11.006). This has been overlooked in the discussion and needs to be added.**

The isoprene discussion has been updated with the reference to Yuan accordingly:

"The significant offsets observed in the PTR-MS data of ~0.1ppb during the afternoon, and ~0.3 ppb during the morning and night, were most likely due to contributions from compounds other than isoprene to the PTR-MS signal at m/z 69……dimethylcyclohexane and cyclopentene in air impacted by vehicle emissions (Yuan et al., 2014). Unfortunately independent measurements of these interferent compounds are not available for this study and their contributions to the PTR-MS signal m/z 69 cannot be estimated."

Note: as shown in the Figure below there is a significant relationship between the difference in PTR-MS and ATVOC measurements and the sum of the cycloalkens measured by the AT-VOC method indicating interference in the signal at m/z

69 in the present study may in part due to contributions from higher cycloalkanes but this effect could not be estimated with the available data.

[Figure]

**The correction of benzene and toluene has been done using literature data, but nothing has been done to confirm the interferences or fragmentations in the PTR-MS used here. The instrument settings influencing the E/N are critical for the fragmentation of larger aromatics to benzene and monoterpenes to toluene. The E/N used here is very low and fragmentation should be reduced compared to the literature data used. Most importantly, the fragmentation in the PTR-MS at a certain setting can be measured very easily by simply exposing the instrument to the pure compound. Using the measured fragmentation patterns and looking at the parent masses 121 or 137 measured during SPS2 should give a much better indication on potential contributions to benzene and toluene and can be used to properly correct mass 79 and 93 in PTR-MS.**

The author's agree this is an optimal approach however these fragmentation tests were not performed at the time of the study. Also, the PTR-MS sensitivity and branching ratio differs for each monoterpene and C9 aromatic isomer but the contribution of each isomer to the total signals at m/z 121 or 137 cannot be determined and as such the AT-VOC data was used.

**If the PTR-MS measurements suffer from interferences, the PTR-MS should be corrected and not the AT-VOC data. You are generating not atmospherically meaningful data. Please subtract from the PTR-MS benzene and toluene signal instead. Also, it needs to be made clear in the figures and tables that corrected data were used. This means changes the formula. Text, table and figures in many places.**

The corrections have been modified accordingly. See revised RMA statistics for Table 5 below.

**Table 5. (REVISED) The (m), intercepts (b) and correlation coefficients ($R^2$) from the RMA regression analysis between the PTR-MS, AT-VOC and DNPH-HPLC measurements. Also included are the estimates of random measurement uncertainty expressed as RMSD for each species and the ratio of the RMSD to the median PTR-MS value expressed as %.**

| m/z | Compound | Slope (m) | Intercept(b) (ppbv) | $R^2$ | RMSD (ppbv) | RMSD/Median | N |
|-----|----------|-----------|---------------------|-------|-------------|-------------|---|
| \multicolumn{8}{l}{[PTR-MS] = m × [AT-VOC] + b} |
| 79 | Benzene (uncorr.) | 1.47 ± 0.04 | 0.02 ± 0.00 | 0.96 | 0.04 | 8 % | 75 |
|    | Benzene (corr.) | 1.39 ± 0.03 | 0.02 ± 0.00 | 0.96 | | | |
| 93 | Toluene (uncorr.) | 1.25 ± 0.02 | -0.03 ± 0.00 | 0.98 | 0.11 | 5 % | 75 |
|    | Toluene (corr.) | 1.21 ± 0.02 | -0.03 ± 0.00 | 0.98 | | | |
| 107 | C$_8$ Aromatics (uncorr.) | 1.16 ± 0.02 | -0.02 ± 0.01 | 0.98 | 0.09 | 7 % | 75 |
|    | C$_8$Aromatics (corr.) | 1.16 ± 0.02 | -0.02 ± 0.01 | 0.98 | | | |
| 69 | Isoprene (all) | 1.23 ± 0.07 | 0.31 ± 0.10 | 0.75 | 0.13 | 25 % | 75 |
|    | Isoprene (5am – 10am) | 1.86 ± 0.32 | 0.28 ± 0.16 | 0.34 | | | |
|    | Isoprene (11am – 7pm) | 1.18 ± 0.06 | 0.11 ± 0.10 | 0.93 | 0.12 | 28% | 26 |
|    | Isoprene (7pm - -5am) | 1.18 ± 0.10 | 0.41 ± 0.33 | 0.83 | | | |
| \multicolumn{8}{l}{[PTR-MS] = m × [DNPH-HPLC] + b} |
| 31 | Formaldehyde | 1.20 ± 0.04 | -0.02 ± 0.02 | 0.90 | 0.16 | 15 % | 53 |
| 45 | Acetaldehyde | 1.43 ± 0.05 | 0.08 ± 0.01 | 0.92 | 0.05 | 6% | 53 |
| 59 | Acetone | 2.01 ± 0.14 | 0.21 ± 0.07 | 0.76 | 0.23 | 15% | 53 |

**The same issues arise in the discussion about DNPH losses. If for example acetone is collected poorly on DNPH, especially at high water mixing ratios, a very simple test with an acetone standard should be done. It seems from the instrument description that only the HPLC was calibrated with a liquid standard, but a calibration should be done**

5 **with the entire sampling setup. This means including the DNPH cartridges. You actually should cross calibrate the PTR-MS standard. From page 5 line 10 it seems that cross calibration was done only with the BTEX standard and the GC-FID PTR-MS combination and also not through the adsorbent tubes.**

The gravimetrically prepared Apel Reimer standard used to calibrate the PTR-MS, was also analysed with the GC-FID-MS

10 against a certified BTEX gas. The FID response factors for the 2 standards differed by 5 − 9% (BTEX/Apel Reimer Ratios: benzene 0.95; toluene 0.95 and m-xylene 0.91) and we can conclude that the PTR-MS and GC-FID-MS calibrations were compatible within these limits.

The authors agree measurements of the automated sampler should be cross calibrated with the certified gaseous standards used to calibrate the PTR-MS for this study, however such tests were not conducted at the time of the study in 2012. As a result of

15 the process undertaken for writing this manuscript these tests have been incorporated into present measurement campaigns but the data is not currently available for use in this manuscript. A list of technical suggestions will be included in the conclusions of the revised manuscript outlining recommendations arising from this study.

**In addition, the DNPH measurements of acetone in Figure 3 actually go to essentially zero at high humidity. In the**

20 **atmosphere, especially in urban areas, so low acetone values have never been observed and clearly point to measurement issues. This should be added as an argument for the ketone loss in the DNPH method.**

See response to comments by referee #2 regarding this issue.

**The selection criteria for measurement comparison #4 of median/MDL>5 should not be used. All the data that are**

25 **above the MDL should be used. That's what MDL means. If you do have a reason not to use those data, you have to**

**change your definition of MDL. The use of these lower concentration data should decrease the R-coef, but the slopes should still be valid.**

The relative error of most measurement systems increases with decreasing VMR (Horwitz, 1982; de Gouw and Warneke, 2007). Using datasets with Median /MDL > 5 was considered suitable for a robust quantitative comparison. While values below the MDL are still included, 50% of the data was > 5 times the MDL ensuring the comparison is not dominated by random instrument noise.

**Minor comments**

**Page 4 line 21: Was the silcosteel inlet tube heated? If not, this is highly recommended.**

The silco-steel inlet tube was not heated, however the automated sampler (DNPH & AT-VOC) and the PTR-MS sampled air via silco steel inlets of almost equal length both connected to a common glass inlet manifold. Consequently, systematic differences due to inlet performance were not expected to be a major factor in the discrepancy between the two methods.

**Page 4 line25: Ellis and Mayhew 2014 ref is missing from the reference list, Lindinger et al 1998 or de Gouw and Warneke 2007 could be mentioned here as well.**

Corrected

**Page 4 line 41: How does your definition of MDL compare to the more standard S/N=2**

The MDL for a single measurement was set at the $95^{th}$ percentile of the deviations about the mean zero. This is approximately equal to an S/N ratio = 2.

**Page 5 line4: Table 2 comes before Table 1 in the text, they should be switched.**

Corrected

**Page 5 line 4: What do you mean with the scatter in the calibration measurements? Was this 10% around the mean for the 30 min daily measurements or are the 10% the deviation for each daily calibration around the campaign average?**

The scatter is the relative std deviation across the campaign average

**Page 5 line 10: Can you add the manufacturers stated uncertainties for the calibration standards?**

Corrected.

**Table 2: The sensitivity of PTR-MS should not be given in ncps/ppb only without indicating the primary ion counts or adding cps/ppb. With ncps/ppb you still don't know the actual instrument sensitivity.**

Table 2 updated. Average H3O+ ion signal was 13.5 million cps.

**Page 5 line 18: "Each compound known to substantially contribute" How do you know which compounds contribute, I guess from literature studies? Please add.**

The m/z signals of the PTR-MS were considered to be dominated by the compound of interest according to the following criteria:

- The compound of interest is known to produce an ion signal at the target m/z in PTR-MS
- A priori knowledge of other VOCs commonly measured in the atmosphere which may be detected at the target m/z in PTR-MS.
- High $R^2$ values in comparison with more selective chromatographic techniques:

    o   The high $R^2$ values of 0.92 give confidence that both the PTR-MS and the DNPH technique were both responding to the same analytes

    o   Acetone samples that were affected by condensation have been removed from the dataset by removing samples obtained when dewpoint was > the automated sampler chiller temp of 7°C used in this study (see response to referee #2 above). Removing this data resulted in an $R^2$ of 0.89 which gives us confidence that the PTR-MS signal at m/z 59 is dominated by acetone.

**Page 5 line 3: What is the difference between RMA and ODR, which is mostly used for instrument comparisons? In ODR the actual instrument uncertainty can be used as the error estimate for both axis. Is that also done in RMA?**

The text will be modified to read "When comparing two observational datasets reduced major axis (RMA) regression, also called geometric mean regression, is preferable to simple least squares linear regression because the analysis is not between an independent and dependent variable, and RMA accounts for random error on both the x- and y- variables, rather than only the y-variable (Kermack and Haldane, 1950; Ayers, 2001; Franq and Govaerts, 2014). The RMA method is recommended when the measurement errors are unknown (Franq and Govaerts, 2014)."

**Table 3: Please indicate the sampling times and PTR-MS averaging times.**

The sampling times and PTR-MS averaging times are presented in Table 1.

**Page 7 line 24: Here and elsewhere in the text: Cappellin et al 2010 is the wrong reference. This should be Cappellin 2012.**

Corrected

**Page 8 line 33: "we correct the AT-VOC data". This should say something like: We calculate the weighted sum of the individual VOCs measured by AT-VOC that corresponds to m/z 107.**

Corrected

**Page 9 line 15: If you separate daytime data, you should also look at night-time data, at night the interference should be the largest.**

Morning and night RMA stats added to revised Table 5 included above. Figures will be updated accordingly.

Briefly, there is no significant correlation between AT-VOC isoprene and the PTR-MS signal at m/z 69 for the period 5am – 10am (R2 = 0.32) rendering the RAM slope and intercept essentially meaningless. There is a slope of 1.9 and offset of 0.41 ppb in the RMA regression for the period 7pm – 11am indicating as suggested by the reviewers that the interference at night and in the early hours of the morning is the largest. The isoprene results and discussion will be updated with these additional details.

**Page 9 line 33: I don't think the influence of other compounds on isoprene by PTR-MS is poorly understood, it is indeed a poorly quantified uncertainty, but we know pretty well what the other compounds are.**

The discussion and conclusions regarding isoprene will be adjusted to reflect that "The influence of these compounds on measurements of isoprene by PTR-MS was poorly quantified in the bottom-up measurement uncertainty analysis of the PTR-MS technique."

**Figure 2: Indicate what the grey squares are. I guess daytime data.**

The grey squares are daytime data and the figure and caption will be adjusted to indicate such.